

# A guideline for reporting experimental protocols in life sciences

Olga Giraldo[1], Alexander Garcia[1,2] and Oscar Corcho[1]

[1] Ontology Engineering Group, Campus de Montegancedo, Boadilla del Monte, Universidad Politécnica de Madrid, Madrid, Spain
[2] Technische Universität Graz, Graz, Austria

## ABSTRACT

Experimental protocols are key when planning, performing and publishing research in many disciplines, especially in relation to the reporting of materials and methods. However, they vary in their content, structure and associated data elements. This article presents a guideline for describing key content for reporting experimental protocols in the domain of life sciences, together with the methodology followed in order to develop such guideline. As part of our work, we propose a checklist that contains 17 data elements that we consider fundamental to facilitate the execution of the protocol. These data elements are formally described in the SMART Protocols ontology. By providing guidance for the key content to be reported, we aim (1) to make it easier for authors to report experimental protocols with necessary and sufficient information that allow others to reproduce an experiment, (2) to promote consistency across laboratories by delivering an adaptable set of data elements, and (3) to make it easier for reviewers and editors to measure the quality of submitted manuscripts against an established criteria. Our checklist focuses on the content, what should be included. Rather than advocating a specific format for protocols in life sciences, the checklist includes a full description of the key data elements that facilitate the execution of the protocol.

## INTRODUCTION

Experimental protocols are fundamental information structures that support the description of the processes by means of which results are generated in experimental research (*Giraldo et al., 2017*; *Freedman, Venugopalan & Wisman, 2017*). Experimental protocols, often as part of "Materials and Methods" in scientific publications, are central for reproducibility; they should include all the necessary information for obtaining consistent results (*Casadevall & Fang, 2010*; *Festing & Altman, 2002*). Although protocols are an important component when reporting experimental activities, their descriptions are often incomplete and vary across publishers and laboratories. For instance, when reporting reagents and equipment, researchers sometimes include catalog numbers and experimental parameters; they may also refer to these items in a generic manner, e.g., "*Dextran sulfate, Sigma-Aldrich*" (*Karlgren et al., 2009*). Having this information is important because reagents usually vary in terms of purity, yield, pH, hydration state,

Corresponding author
Olga Giraldo, ogiraldo@fi.upm.es

grade, and possibly additional biochemical or biophysical features. Similarly, experimental protocols often include ambiguities such as "*Store the samples at room temperature until sample digestion*" (*Brandenburg et al., 2002*); but, how many Celsius degrees? What is the estimated time for digesting the sample? Having this information available not only saves time and effort, it also makes it easier for researchers to reproduce experimental results; adequate and comprehensive reporting facilitates reproducibility (*Freedman, Venugopalan & Wisman, 2017*; *Baker, 2016*).

Several efforts focus on building data storage infrastructures, e.g., 3TU. Datacentrum (*4TU, 2017*), CSIRO Data Access Portal (*CSIRO, 2017*), Dryad (*Dryad, 2017*), figshare (*Figshare, 2017*), Dataverse (*King, 2007*) and Zenodo (*Zenodo, 2017*). These data repositories make it possible to review the data and evaluate whether the analysis and conclusions drawn are accurate. However, they do little to validate the quality and accuracy of the data itself. Evaluating research implies being able to obtain similar, if not identical results. Journals and funders are now asking for datasets to be publicly available for reuse and validation. Fully meeting this goal requires datasets to be endowed with auxiliary data providing contextual information e.g., methods used to derive such data (*Assante et al., 2016*; *Simmhan, Plale & Gannon, 2005*). If data must be public and available, shouldn't methods be equally public and available?

Illustrating the problem of adequate reporting, *Moher et al. (2015)* have pointed out that fewer than 20% of highly-cited publications have adequate descriptions of study design and analytic methods. In a similar vein, *Vasilevsky et al. (2013)* showed that 54% of biomedical research resources such as model organisms, antibodies, knockdown reagents (morpholinos or RNAi), constructs, and cell lines are not uniquely identifiable in the biomedical literature, regardless of journal Impact Factor. Accurate and comprehensive documentation for experimental activities is critical for patenting, as well as in cases of scientific misconduct. Having data available is important; knowing how the data were produced is just as important. Part of the problem lies in the heterogeneity of reporting structures; these may vary across laboratories in the same domain. Despite this variability, we want to know which data elements are common and uncommon across protocols; we use these elements as the basis for suggesting our guideline for reporting protocols. We have analyzed over 500 published and non-published experimental protocols, as well as guidelines for authors from journals publishing protocols. From this analysis we have derived a practical adaptable checklist for reporting experimental protocols.

Efforts such as the Structured, Transparent, Accessible Reporting (STAR) initiative (*Marcus, 2016*; *Cell Press, 2017*) address the problem of structure and standardization when reporting methods. In a similar manner, The Minimum Information about a Cellular Assay (MIACA) (*MIACA, 2017*), The Minimum Information about a Flow Cytometry Experiment (MIFlowCyt) (*Lee et al., 2008*) and many other "*minimal information*" efforts deliver minimal data elements describing specific types of experiments. *Soldatova et al. (2008)* and *Soldatova et al. (2014)* proposes the EXACT ontology for representing experimental actions in experimental protocols; similarly, *Giraldo et al. (2017)* proposes the **SeMA**ntic **R**epresen**T**ation of Protocols ontology (henceforth SMART Protocols Ontology) an ontology for reporting experimental protocols and the corresponding workflows. These

approaches are not minimal; they aim to be comprehensive in the description of the workflow, parameters, sample, instruments, reagents, hints, troubleshooting, and all the data elements that help to reproduce an experiment and describe experimental actions.

There are also complementary efforts addressing the problem of identifiers for reagents and equipment; for instance, the Resource Identification Initiative (RII) (*Force11, 2017*), aims to help researchers sufficiently cite the key resources used to produce the scientific findings. In a similar vein, the Global Unique Device Identification Database (GUDID) (*NIH, 2018*) has key device identification information for medical devices that have Unique Device Identifiers (UDI); the Antibody Registry (*Antibody Registry, 2018*), gives researchers a way to universally identify antibodies used in their research, and also the Addgene web-application (*Addgene, 2018*) makes it easy for researchers to identify plasmids. Having identifiers make it possible for researchers to be more accurate in their reporting by unequivocally pointing to the resource used or produced. The Resource Identification Portal (*RIP, 2018*), makes it easier to navigate through available identifiers, researchers can search across all the sources from a single location.

In this paper, we present a guideline for reporting experimental protocols; we complement our guideline with a machine-processable checklist that helps researchers, reviewers and editors to measure the completeness of a protocol. Each data element in our guideline is represented in the SMART Protocols Ontology. This paper is organized as follows: we start by describing the materials and methods used to derive the resulting guidelines. In the "Results" section, we present examples indicating how to report each data element; a machine readable checklist in the JavaScript Object Notation (JSON) format is also presented in this section. We then discuss our work and present the conclusions.

## MATERIALS AND METHODS

### Materials

We have analyzed: (i) guidelines for authors from journals publishing protocols (*Giraldo, Garcia & Corcho, 2018b*), (ii) our corpus of protocols (*Giraldo, Garcia & Corcho, 2018a*), (iii) a set of reporting structures proposed by minimal information projects available in the FairSharing catalog (*McQuilton et al., 2016*), and (iv) relevant biomedical ontologies available in BioPortal (*Whetzel et al., 2011*) and Ontobee (*Xiang et al., 2011*). Our analysis was carried out by a domain expert, Olga Giraldo; she is an expert in text mining and biomedical ontologies with over ten years of experience in laboratory techniques. All the documents were read, and then data elements, subject areas, materials (e.g., sample, kits, solutions, reagents, etc.), and workflow information were identified. Resulting from this activity we established a baseline terminology, common and non common data elements, as well as patterns in the description of the workflows (e.g., information describing the steps and the order for the execution of the workflow).

### *Instructions for authors from analyzed journals*

Publishers usually have instructions for prospective authors; these indications tell authors what to include, the information that should be provided, and how it should be reported in the manuscript. In Table 1 we present the list of guidelines that were analyzed.

**Table 1** Guidelines for reporting experimental protocols.

| Journal | Guidelines for authors |
|---------|------------------------|
| BioTechniques (BioTech) | *Giraldo, Garcia & Corcho (2018b)* |
| CSH protocols (CSH) | *Cold Spring Harbor Press (2013)* |
| Current Protocols (CP) | *Wiley's Current Protocols (2012)* |
| Journal of Visualized Experiments (JoVE) | *JoVE (2012)* |
| Nature Protocols (NP) | *Nature Protocols (2012)* |
| Springer Protocols (SP) | *Springer Protocols (2013)* |
| MethodsX | *MethodsX (2014)* |
| Bio-protocols (BP) | *Bio-protocol (2012)* |
| Journal of Biological Methods (JBM) | *JBM (2013)* |

**Table 2** Corpus of protocols analyzed.

| Source | Number of protocols |
|--------|---------------------|
| BioTechniques (BioTech) | 16 |
| CSH protocols (CSH) | 267 |
| Current Protocols (CP) | 31 |
| Genetics and Molecular Research (GMR) | 5 |
| Journal of Visualized Experiments (JoVE) | 21 |
| Nature Protocols Exchange (NPE) | 39 |
| Plant Methods (PM) | 12 |
| Plos One (PO) | 5 |
| Springer Protocols (SP) | 5 |
| MethodsX | 7 |
| Bio-protocols (BP) | 40 |
| Journal of Biological Methods (JBM) | 7 |
| Non-published protocols from CIAT | 75 |

### Corpus of protocols

Our corpus includes 530 published and unpublished protocols. Unpublished protocols (75 in total) were collected from four laboratories located at the International Center for Tropical Agriculture (CIAT) (*CIAT, 2017*). The published protocols (455 in total) were gathered from the repository "Nature Protocol Exchange" (*NPE, 2017*) and from 11 journals, namely: BioTechniques, Cold Spring Harbor Protocols, Current Protocols, Genetics and Molecular Research (*GMR, 2017*), JoVE, Plant Methods (*BioMed Central, 2017*), Plos One (*PLOS ONE, 2017*), Springer Protocols, MethodsX, Bio-Protocol and the Journal of Biological Methods. The analyzed protocols comprise areas such as cell biology, molecular biology, immunology, and virology. The number of protocols from each journal is presented in Table 2.

### Minimum information standards and ontologies

We analyzed minimum information standards from the FairSharing catalog, e.g., MIAPPE (*MIAPPE, 2017*), MIARE (*MIARE, 2017*) and MIQE (*Bustin et al., 2009*). See Table 3 for the complete list of minimum information models that we analyzed.

**Table 3   Minimum information standards analyzed.**

| Standards | Description |
|---|---|
| Minimum Information about Plant Phenotyping Experiment (MIAPPE) | A reporting guideline for plant phenotyping experiments. |
| CIMR: Plant Biology Context (*Nikolau et al., 2006*) | A standard for reporting metabolomics experiments. |
| The Gel Electrophoresis Markup Language (GelML) | A standard for representing gel electrophoresis experiments performed in proteomics investigations. |
| Minimum Information about a Cellular Assay (MIACA) | A standardized description of cell-based functional assay projects. |
| Minimum Information About an RNAi Experiment (MIARE) | A checklist describing the information that should be reported for an RNA interference experiment. |
| The Minimum Information about a Flow Cytometry Experiment (MIFlowCyt) | This guideline describes the minimum information required to report flow cytometry (FCM) experiments. |
| Minimum Information for Publication of Quantitative Real-Time PCR Experiments (MIQE) | This guideline describes the minimum information necessary for evaluating qPCR experiments. |
| ARRIVE (Animal Research: Reporting of *In Vivo* Experiments) (*Kilkenny et al., 2010*) | Initiative to improve the standard of reporting of research using animals. |

We paid special attention to the recommendations indicating how to describe specimens, reagents, instruments, software and other entities participating in different types of experiments. Ontologies available at Bioportal and Ontobee were also considered; we focused on ontologies modeling domains, e.g., bioassays (BAO), protocols (EXACT), experiments and investigations (OBI). We also focused on those modeling specific entities, e.g., organisms (NCBI Taxon), anatomical parts (UBERON), reagents or chemical compounds (ERO, ChEBI), instruments (OBI, BAO, EFO). The list of analyzed ontologies is presented in Table 4.

## Methods for developing this guideline

Developing the guideline entailed a series of activities; these were organized in the following stages: (i) analysis of guidelines for authors, (ii) analysis of protocols, (iii) analysis of Minimum Information (MI) standards and ontologies, and (iv) evaluation of the data elements from our guideline. For a detailed representation of our workflow, see Fig. 1

### Analyzing guidelines for authors

We manually reviewed instructions for authors from nine journals as presented in Table 1. In this stage (step A in Fig. 1), we identified bibliographic data elements classified as "desirable information" in the analyzed guidelines. See Table 5.

In addition, we identified the rhetorical elements. These have been categorized in the guidelines for authors as: (i) required information (R), must be submitted with the manuscript; (ii) desirable information (D), should be submitted if available; and (iii) optional (O) or extra information. See Table 6 for more details.

**Table 4  Ontologies analyzed.**

| Ontology | Description |
|---|---|
| The Ontology for Biomedical Investigations (OBI) (*Bandrowski et al., 2016*) | An ontology for the description of life-science and clinical investigations. |
| The Information Artifact Ontology (IAO) (*IAO, 2017*) | An ontology of information entities. |
| The ontology of experiments (EXPO) (*Soldatova & King, 2006*) | An ontology about scientific experiments. |
| The ontology of experimental actions (EXACT) | An ontology representing experimental actions. |
| The BioAssay Ontology (BAO) (*Abeyruwan et al., 2014*) | An ontology describing biological assays. |
| The Experimental Factor Ontology (EFO) (*Malone et al., 2010*) | The ontology includes aspects of disease, anatomy, cell type, cell lines, chemical compounds and assay information. |
| eagle-i resource ontology (ERO) | An ontology of research resources such as instruments, protocols, reagents, animal models and biospecimens. |
| NCBI taxonomy (NCBITaxon) (*Federhen, 2015*) | An ontology representation of the NCBI organismal taxonomy. |
| Chemical Entities of Biological Interest (ChEBI) (*Hastings et al., 2013*) | Classification of molecular entities of biological interest focusing on 'small' chemical compounds. |
| Uberon multi-species anatomy ontology (UBERON) (*Mungall et al., 2012*) | A cross-species anatomy ontology covering animals and bridging multiple species-specific ontologies. |
| Cell Line Ontology (CLO) (*Sarntivijai et al., 2014*; *Sarntivijai et al., 2011*) | The ontology was developed to standardize and integrate cell line information. |

**Table 5  Bibliographic data elements from guidelines for authors.**

| Bibliographic data elements | BioTech | NP | CP | JoVE | CSH | SP | BP | MethodsX | JBM |
|---|---|---|---|---|---|---|---|---|---|
| title/name | Y | Y | Y | Y | Y | Y | Y | Y | Y |
| author name | Y | Y | Y | Y | Y | Y | Y | Y | Y |
| author identifier (e.g., orcid) | N | N | N | N | N | N | N | N | N |
| protocol identifier (DOI) | Y | Y | Y | Y | Y | Y | Y | Y | Y |
| protocol source (retrieved from, modified from) | N | Y | N | N | N | N | N | N | N |
| updates (corrections, retractions or other revisions) | N | N | N | N | N | N | N | N | N |
| references/related publications | Y | Y | Y | Y | Y | Y | Y | Y | Y |
| categories or keywords | Y | Y | Y | Y | Y | Y | Y | Y | Y |

**Notes.**

Y, datum considered as "desirable information" if this is available; N, datum not considered in the guidelines.

### Analyzing the protocols

In 2014, we started by manually reviewing 175 published and unpublished protocols; these were from domains such as cell biology, biotechnology, virology, biochemistry and pathology. From this collection, 75 are unpublished protocols and thus not available in the dataset for this paper. These unpublished protocols were collected from four laboratories located at the CIAT. In 2015, our corpus grew to 530; we included 355 published protocols

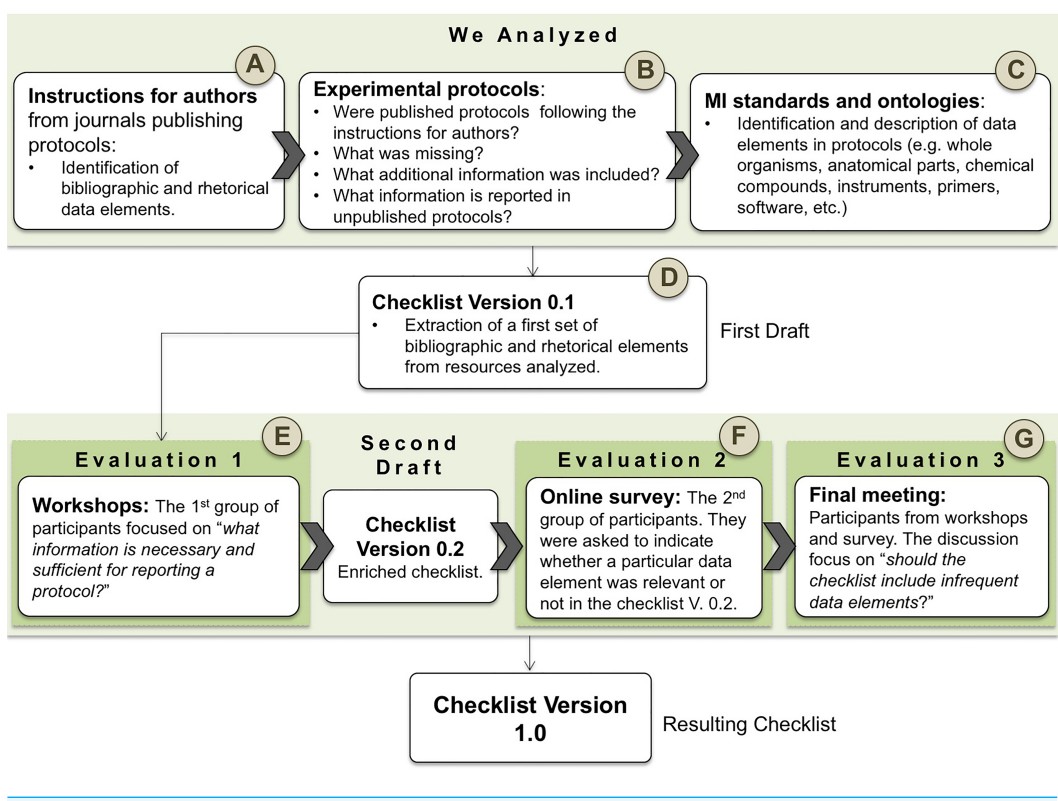

**Figure 1  Methodology workflow.**

gathered from one repository and eleven journals as listed in Table 2. Our corpus of published protocols is: (i) identifiable, i.e., each document has a Digital Object Identifier (DOI) and (ii) in disciplines and areas related to the expertise provided by our domain experts, e.g., virology, pathology, biochemistry, biotechnology, plant biotechnology, cell biology, molecular and developmental biology and microbiology. In this stage, step B in Fig. 1, we analyzed the content of the protocols; theory vs. practice was our main concern. We manually verified if published protocols were following the guidelines; if not, *what was missing, what additional information was included?* We also reviewed common data elements in unpublished protocols.

### Analyzing minimum information standards and ontologies

Biomedical sciences have an extensive body of work related to minimum information standards and reporting structures, e.g., those from the FairSharing initiative. We were interested in determining whether there was any relation to these resources. Our checklist includes the data elements that are common across these resources. We manually analyzed standards such as MIQE, used to describe qPCR assays; we also looked into MIACA, it provides guidelines to report cellular assays; ARRIVE, which provides detailed descriptions of experiments on animal models and MIAPPE, addressing the descriptions of experiments for plant phenotyping. See Table 3 for a complete list of the standards that we analyzed. Metadata, data, and reporting structures in biomedical documents are frequently related to

**Table 6  Rhetorical/Discourse elements from guidelines for authors.**

| Rhetorical/discourse elements | Bio-Tech | NP | CP | JoVE | CSH | SP | BP | MethodsX | JBM |
|---|---|---|---|---|---|---|---|---|---|
| Description of the protocol (objective, range of applications where the protocol can be used, advantages, limitations) | D | D | D | D | D | D | D | D | D |
| Description of the sample tested (name; ID; strain, line or ecotype; developmental stage; organism part; growth conditions; treatment type; size) | NC | NC | D | NC | NC | NC | NC | NC | NC |
| Reagents (name, vendor, catalog number) | R | D | D | D | R | D | R | NC | D |
| Equipment (name, vendor, catalog number) | R | D | D | D | R | D | R | NC | D |
| Recipes for solutions (name, final concentration, volume) | R | D | D | D | D | D | R | NC | D |
| Procedure description | R | R | R | D | R | R | R | R | D |
| Alternatives to performing specific steps | NC | NC | D | D | NC | D | NC | NC | NC |
| Critical steps | R | NC | D | NC | NC | NC | NC | NC | NC |
| Pause point | R | NC | NC | O | D | NC | NC | NC | NC |
| Troubleshooting | R | O | R | O | D | D | NC | NC | D |
| Caution/warnings | NC | NC | R | O | NC | D | NC | NC | D |
| Execution time | NC | O | D | NC | NC | D | NC | NC | NC |
| Storage conditions (reagents, recipes, samples) | R | NC | R | D | D | D | NC | NC | NC |
| Results (figure, tables) | R | NC | R | R | D | R | D | NC | D |

Notes.

R, Required information; NC, Not Considered in guidelines; D, Desirable information; O, Optional information.

ontological concepts. We also looked into relations between data elements and biomedical ontologies available in BioPortal and Ontobee. We focused on ontologies representing materials that are often found in protocols; for instance, organisms, anatomical parts (e.g., CLO, UBERON, NCBI Taxon), reagents or chemical compounds (e.g., ChEBI, ERO), and equipment (e.g., OBI, BAO, EFO). The complete list of the ontologies that we analyzed is presented in Table 4.

### Generating the first draft

The first draft is the main output from the initial analysis of instructions for authors, experimental protocols, MI standards and ontologies, see (step D in Fig. 1). The data elements were organized into four categories: bibliographic data elements such as title, authors; descriptive data elements such as purpose, application; data elements for materials, e.g., sample, reagents, equipment; and data elements for procedures, e.g., critical steps, Troubleshooting. The role of the authors, provenance and properties describing the sample (e.g., organism part, amount of the sample, etc.) were considered in this first draft. In addition properties like "name", "manufacturer or vendor" and "identifier" were proposed to describe equipment, reagents and kits.

### Evaluation of data elements by domain experts

This stage entailed three activities. The first activity was carried out at CIAT with the participation of 19 domain experts in areas such as virology, pathology, biochemistry, and plant biotechnology. The input of this activity was the checklist V. 0.1 (see step E in Fig. 1).

This evaluation focused on "*What information is necessary and sufficient for reporting an experimental protocol?*"; the discussion also addressed data elements that were not initially part of guidelines for authors -e.g., consumables. The result of this activity was the version 0.2 of the checklist; domain experts suggested to use an online survey for further validation. This survey was designed to enrich and validate the checklist V. 0.2. We used a Google survey that was circulated over mailing lists; participants did not have to disclose their identity (see step F in Fig. 1). A final meeting was organized with those who participated in workshops, as well as in the survey (23 in total) to discuss the results of the online poll. The discussion focused on the question: *Should the checklist include data elements not considered by the majority of participants?* Participants were presented with use cases where infrequent data elements are relevant in their working areas. It was decided to include all infrequent data elements; domain experts concluded that this guideline was a comprehensive checklist a opposed to a minimal information. Also, after discussing infrequent data elements it was concluded that the importance of a data element should not bear a direct relation to its popularity. The analogy used was that of an editorial council; some data elements needed to be included regardless of the popularity as an editorial decision. The output of this activity was the checklist V. 1.0. The survey and its responses are available at (*Giraldo, Garcia & Corcho, 2018c*). This current version includes a new bibliographic element "license of the protocol", as well as the property "equipment configuration" associated to the datum equipment. The properties: alternative, optional and parallel steps were added to describe the procedure. In addition, the datum "PCR primers" was removed from the checklist, it is specific and therefore should be the product of a community specialization as opposed to part of a generic guideline.

## RESULTS

Our results are summarized in Table 7; it includes all the data elements resulting from the process illustrated in Fig. 1. We have also implemented our checklist as an online tool that generates data in the JSON format and presents an indicator of completeness based on the checked data elements; the tool is available at https://smartprotocols.github.io/checklist1.0 (*Gómez, alexander & Giraldo, 2018*). Below, we present a complete description of the data elements in our checklist. We have organized the data elements in four categories, namely: (i) bibliographic data elements, (ii) discourse data elements, (iii) data elements for materials, and iv) data elements for the procedure. Ours is a comprehensive checklist, the data elements must be reported whenever applicable.

### Bibliographic data elements

From the guidelines for authors, the datum "author identifier" was not considered, nor was this data element found in the analyzed protocols. The "provenance" is proposed as "desirable information" in only two of the guidelines (Nature Protocols and Bio-protocols), as well as "updates of the protocol" (Cold Spring Harbor Protocols and Bio-protocols). A total of 72.5% (29) of the protocols available in our Bio-protocols collection and 61.5% (24) of the protocols available in our Nature Protocols Exchange collection reported the

**Table 7  Data elements for reporting protocols in life sciences.**

| Data element | Property |
|---|---|
| Title of the protocol | |
| Author | Name |
| | Identifier |
| Version number | |
| License of the protocol | |
| Provenance of the protocol | |
| Overall objective or purpose | |
| Application of the protocol | |
| Advantage(s) of the protocol | |
| Limitation(s) of the protocol | |
| Organism | Whole organism / Organism part |
| | Sample/organism identifier |
| | Strain, genotype or line |
| | Amount of Bio-Source |
| | Developmental stage |
| | Bio-source supplier |
| | Growth substrates |
| | Growth environment |
| | Growth time |
| | Sample pre-treatment or sample preparation |
| Laboratory equipment | Name |
| | Manufacturer or vendor (including homepage) |
| | Identifier (catalog number or model) |
| | Equipment configuration |
| Laboratory consumable | Name |
| | Manufacturer or vendor (including homepage) |
| | Identifier (catalog number) |
| Reagent | Name |
| | Manufacturer or vendor (including homepage) |
| | Identifier (catalog number) |
| Kit | Name |
| | Manufacturer or vendor (including homepage) |
| | Identifier (catalog number) |
| Recipe for solution | Name |
| | Reagent or chemical compound name |
| | Initial concentration of a chemical compound |
| | Final concentration of chemical compound |
| | Storage conditions |
| | Cautions |
| | Hints |

**Table 7** (*continued*)

| Data element | Property |
|---|---|
| Software | Name |
| | Version number |
| | Homepage |
| Procedure | List of steps in numerical order |
| | Alternative/Optional/Parallel steps |
| | Critical steps |
| | Pause point |
| | Timing |
| | Hints |
| | Troubleshooting |

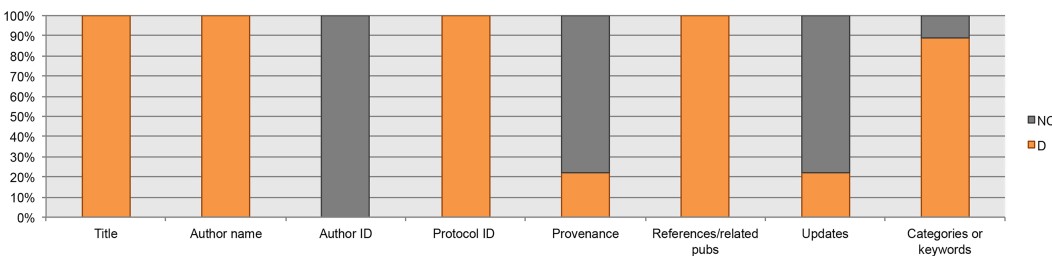

**Figure 2** **Bibliographic data elements found in guidelines for authors.** NC, Not Considered in guidelines; D, Desirable information if this is available.

provenance (Fig. 2). None of the protocols collected from Cold Spring Harbor Protocols or Bio-protocols had been updated–last checked December 2017.

As a result of the workshops, domain experts exposed the importance of including these three data elements in our checklist. For instance, readers sometimes need to contact the authors to ask about specific information (quantity of the sample used, the storage conditions of a solution prepared in the lab, etc.); occasionally, the correspondent author does not respond because he/she has changed his/her email address, and searching for the full name could retrieve multiple results. By using author IDs, this situation could be resolved. The experts asserted that well-documented provenance helps them to know where the protocol comes from and whether it has changed. For example, domain experts expressed their interest in knowing where a particular protocol was published for the first time, who has reused it, how many research papers have used it, how many people have modified it, etc. In a similar way, domain experts also expressed the need for a version control system that could help them to know and understand how, where and why the protocol has changed. For example, researchers are interested in tracking changes in quantities, reagents, instruments, hints, etc. For a complete description of the bibliographic data elements proposed in our checklist, see below.

*Title.* The title should be informative, explicit, and concise (50 words or fewer). The use of ambiguous terminology and trivial adjectives or adverbs (e.g., novel, rapid,

| Table 8 | Examples illustrating two tittles. | |
|---|---|---|
| ambiguous title | A **single**[a] protocol for extraction of **gDNA**[b] from bacteria and yeast. | Protocol available at *Vingataramin & Frost (2015)* |
| comprehensible title | Extraction of nucleic acids from yeast cells and plant tissues using ethanol as medium for sample preservation and cell disruption. | Protocol available at *Linke et al. (2010)* |

**Notes.**
Issues in the ambiguous tittle:
[a] Use of ambiguous terminology.
[b] use of abbreviations.

efficient, inexpensive, or their synonyms) should be avoided. The use of numerical values, abbreviations, acronyms, and trademarked or copyrighted product names is discouraged. This definition was adapted from BioTechniques (*Giraldo, Garcia & Corcho, 2018b*). In Table 8, we present examples illustrating how to define the title.

*Author name and author identifier.* The full name(s) of the author(s) is required together with an author ID, e.g., ORCID (*ORCID, 2017*) or research ID (*ResearcherID, 2017*). The role of each author is also required; depending on the domain, there may be several roles. It is important to use a simple word that describes who did what. Publishers, laboratories, and authors should enforce the use of an "author contribution section" to identify the role of each author. We have identified two roles that are common across our corpus of documents.

- *Creator of the protocol:* This is the person or team responsible for the development or adaptation of a protocol.
- *Laboratory-validation scientist:* Protocols should be validated in order to certify that the processes are clearly described; it must be possible for others to follow the described processes. If applicable, statistical validation should also be addressed. The validation may be procedural (related to the process) or statistical (related to the statistics). According to the Food and Drug Administration (FDA) (*FDA, 2017*), validation is "*establishing documented evidence which provides a high degree of assurance that a specific process will consistently produce a product meeting its predetermined specifications and quality attributes*" (*Das, 2011*).

*Updating the protocol.* The peer-reviewed and non peer-reviewed repositories of protocols should encourage authors to submit updated versions of their protocols; these may be corrections, retractions, or other revisions. Extensive modifications to existing protocols could be published as adapted versions and should be linked to the original protocol. We recommended to promote the use of a version control system; in this paper we suggest to use the version control guidelines proposed by the National Institute of Health (NIH) (*NIH, 2017*).

- *Document dates:* Suitable for unpublished protocols. The date indicating when the protocol was generated should be in the first page and, whenever possible, incorporated into the header or footer of each page in the document.

| Table 9 | Example illustrating the provenance of a protocol. | |
|---|---|---|
| example | *"This protocol was adapted from "How to Study Gene Expression," Chapter 7, in Arabidopsis: A Laboratory Manual (eds. Weigel and Glazebrook). Cold Spring Harbor Laboratory Press, Cold Spring Harbor, NY, USA, 2002."* | Protocol available at *Blazquez (2007)* |

- *Version numbers:* Suitable for unpublished protocols. The current version number of the protocol is identified in the first page and, when possible, incorporated into the header or footer of each page of the document.

  - *Draft document version number:* Suitable for unpublished protocols. The first draft of a document will be Version 0.1. Subsequent drafts will have an increase of "0.1" in the version number, e.g., 0.2, 0.3, 0.4, ... 0.9, 0.10, 0.11.
  - *Final document version number and date:* Suitable for unpublished and published protocols. The author (or investigator) will deem a protocol final after all reviewers have provided final comments and these have been addressed. The first final version of a document will be Version 1.0; the date when the document becomes final should also be included. Subsequent final documents will have an increase of "1.0" in the version number (1.0, 2.0, etc.).

- *Documenting substantive changes:* Suitable for unpublished and published protocols. A list of changes from the previous drafts or final documents will be kept. The list will be cumulative and identify the changes from the preceding document versions so that the evolution of the document can be seen. The list of changes and consent/assent documents should be kept with the final protocol.

*Provenance of the protocol.* The provenance is used to indicate whether or not the protocol results from modifying a previous one. The provenance also indicates whether the protocol comes from a repository, e.g., Nature Protocols Exchange, protocols.io (*Teytelman et al., 2016*), or a journal like JoVE, MethodsX, or Bio-Protocols. The former refers to adaptations of the protocol. The latter indicates where the protocol comes from. See Table 9.

*License of the protocol.* The protocols should include a license. Whether as part of a publication or, just as an internal document, researchers share, adapt and reuse protocols. The terms of the license should facilitate and make clear the legal framework for these activities.

## Data elements of the discourse

Here, we present the elements considered necessary to understand the suitability of a protocol. They are the "overall objective or purpose", "applications", "advantages," and "limitations". 100% of the analyzed guidelines for author suggest the inclusion of these four elements in the abstract or introduction section. However, one or more of these four elements were not reported. For example, "limitations" was reported in only 20% of the protocols from Genetic and Molecular Research and PLOS One, and in 40% of the protocols from Springer. See Fig. 3.

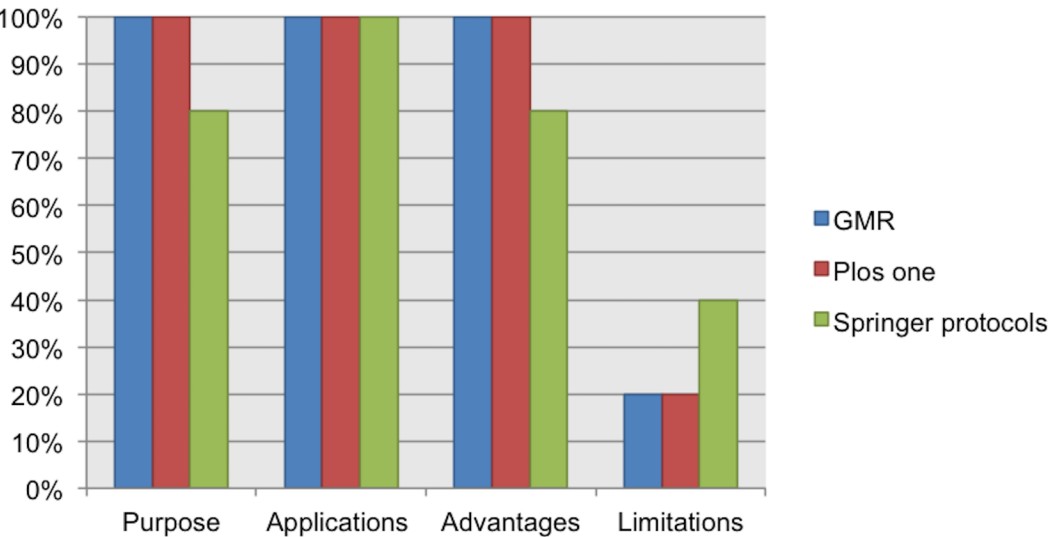

**Figure 3** **Data elements related to the discourse as reported in the analyzed protocols.**

Interestingly, 83% of the respondents considered the "limitations" to be a data element that is necessary when reporting a protocol. In the last meeting, participants considered that "limitations" represents an opportunity to make suggestions for further improvements. Another data element discussed was "advantages"; 43% of the respondents considered the "advantages" as a data element that is necessary to be reported in a protocol. In the last meeting, all participants agreed that "advantages" (where applicable) could help us to compare a protocol with other alternatives commonly used to achieve the same result. For a complete description of the discourse data elements proposed in our checklist, see below.

***Overall objective or Purpose.*** The description of the objective should make it possible for readers to decide on the suitability of the protocol for their experimental problem. See Table 10.

***Application of the protocol.*** This information should indicate the range of techniques where the protocol could be applied. See Table 10.

***Advantage(s) of the protocol.*** Here, the advantages of a protocol compared to other alternatives should be discussed. See Table 10. Where applicable, references should be made to alternative methods that are commonly used to achieve the same result.

***Limitation(s) of the protocol.*** This datum includes a discussion of the limitations of the protocol. This should also indicate the situations in which the protocol could be unreliable or unsuccessful. See Table 10.

## Data elements for materials

From the analyzed guidelines for authors, the datum "sample description" was considered only in the Current Protocols guidelines. The "laboratory consumables or supplies" datum was not included in any of the analyzed guidelines. See Fig. 4.

**Table 10   Examples of discursive data elements.**

| Discourse data element | Example | Source |
|---|---|---|
| Overall objective/ Purpose | "*Development of a method to isolate small RNAs from different plant species (…) that no need of first total RNA extraction and is not based on the commercially available TRIzol® Reagent or columns.*" | Protocol available at *Rosas-Cárdenas et al. (2011)* |
| Application | "*DNA from this experiment can be used for all kinds of genetics studies, including genotyping and mapping.*" | Protocol available at *Lu (2011)* |
| Advantage(s) | "*We describe a fast, efficient and economic in-house protocol for plasmid preparation using glass syringe filters. Plasmid yield and quality as determined by enzyme digestion and transfection efficiency were equivalent to the expensive commercial kits. Importantly, the time required for purification was much less than that required using a commercial kit.*" | Protocol available at *Kim & Morrison (2009)* |
| Limitation(s) | "*A major problem faced both in this and other safflower transformation studies is the hyperhydration of transgenic shoots which result in the loss of a large proportion of transgenic shoots.*" | Protocol available at *Belide et al. (2011)* |

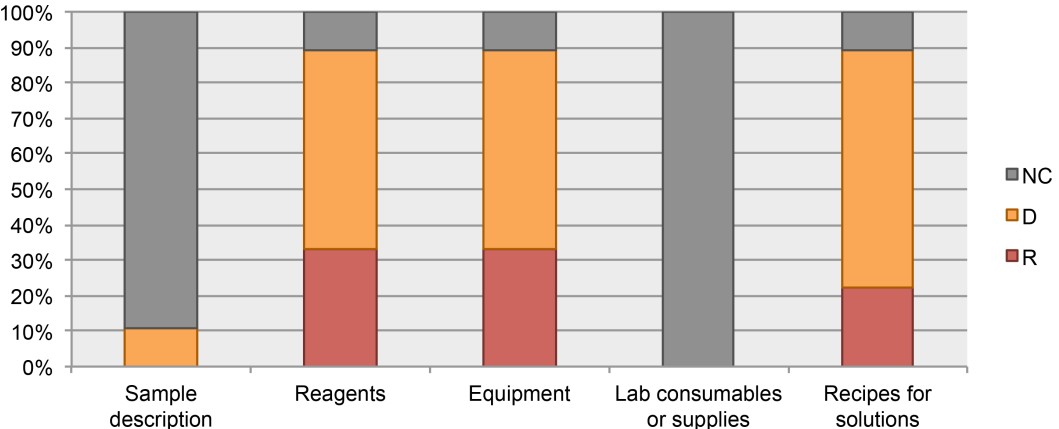

**Figure 4   Data elements describing materials.** NC, Not Considered in guidelines; D, Desirable information if this is available; R, Required information.

Our Current Protocols collection includes documents about toxicology, microbiology, magnetic resonance imaging, cytometry, chemistry, cell biology, human genetics, neuroscience, immunology, pharmacology, protein, and biochemistry; for these protocols the input is a biological or biochemical sample. This collection also includes protocols in bioinformatics with data as the input. 100% of the protocols from our Current Protocols collection includes information about the input of the protocol (biological/biochemical sample or data). In addition, 87% of protocols from this collection include a list of materials or resources (reagents, equipment, consumables, software, etc.).
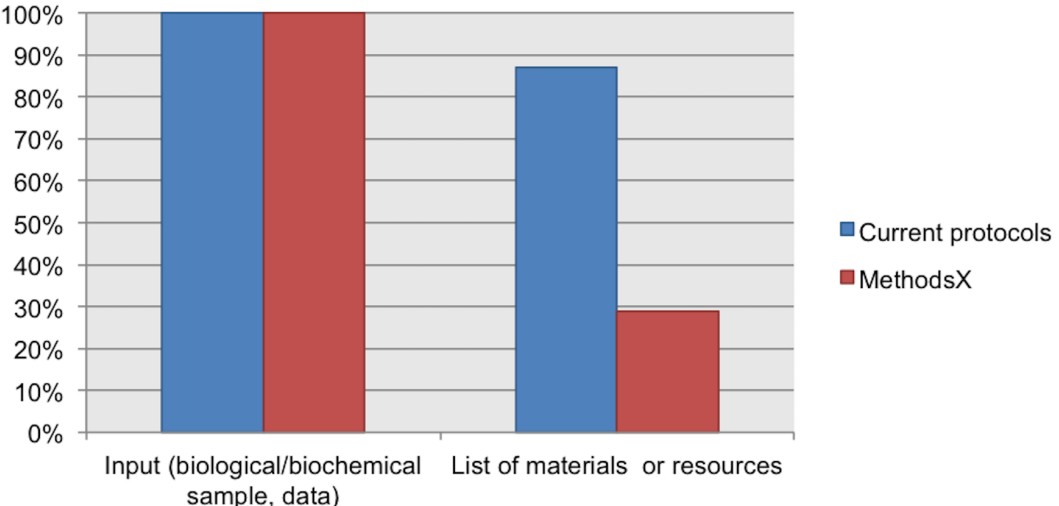

Figure 5    **Data elements describing materials.**

We also analyzed the protocols from our MethodsX collection. We found that despite the exclusion of the sample description in guidelines for authors, the authors included this information in their protocols. Unfortunately, these protocols do not include a list of materials. Only 29% of the protocols reported a partial list of materials. For example, the protocol published by *Vingataramin & Frost (2015)*, includes a list of recommended equipment but does not list any of the reagents, consumables, or other resources mentioned in the protocol instructions. See Fig. 5.

Domain experts considered that the input of the protocol (biological/biochemical sample or data) needs an accurate description; the granularity of the description varies depending on the domain. If such description is not available then the reproducibility could be affected. In addition, domain experts strongly suggested to include consumables in the checklist. It was a general surprise not to find these data elements in the guidelines for authors that we analyzed. Domain experts shared with us bad experiences caused by the lack of information about the type of consumables. Some of the incidents that may arise from the lack of this information include: (i) cross contamination, when no information suggesting the use of filtered pipet tips is available; (ii) misuse of containers, when no information about the use of containers resistant to extreme temperatures and/or impacts is available; (iii) misuse of containers, when a container made of a specific material should be used, e.g., glass vs. plastic vs. metal. This is critical information; researchers need to know if reagents or solutions prepared in the laboratory require some specific type of containers in order to avoid unnecessary reactions altering the result of the assay. Presented below is the set of data elements related to materials or resources used for carrying out the execution of a protocol.

**Sample.** This is the role played by a biological substance; the sample is an experimental input to a protocol. The information required depends on the type of sample being

described and the requirements from different communities. Here, we present the data elements for samples commonly used across the protocols and guidelines that we analyzed.

- **Bio-source properties:**

  **Strain, genotype or line:** This datum is about subspecies such as ecotype, cultivar, accession, or line. In the case of crosses or breeding results, pedigree information should also be provided.

  **Starting material:** This datum is about the physical biological specimen from which your experimental data are derived. The starting material could be a whole organism, or a part of this.

  - **whole organism** Typical examples are multicellular animals, plants, and fungi; or unicellular microorganisms such as a protists, bacteria, and archaea.
  - **organism part** Typical examples of an organism part include a cell line, a tissue, an organ, corporal bodily fluids protoplasts, nucleic acids, proteins, etc.
  - **organism/sample identifier** This is the unique identifier assigned to an organism. The NCBI taxonomy id, also known as "taxid", is commonly used to identify an organism; the Taxonomy Database is a curated classification and nomenclature for all organisms in the public sequence databases. Public identification systems, e.g., the Taxonomy Database, should be used when ever possible. Identifiers may be internal; for instance, laboratories often have their own coding system for generating identifiers. When reporting internal identifiers it is important to also state the source and the nature (private or pubic) of the identifier, e.g., A0928873874, barcode (CIAT-DAPA internal identifier) of a specimen or sample.

  **Amount of Bio-Source:** This datum is about mass (mg fresh weight or mg dry weight), number of cells, or other measurable bulk numbers (e.g., protein content).

  **Developmental stage:** This datum includes age and gender (if applicable) of the organism.

  **Bio-source Supplier:** This datum is defined as a person, company, laboratory or entity that offers a variety of biosamples or biospecimens.

- **Growth conditions:**

  **Growth substrates:** This datum refers to an hydroponic system (type, supplier, nutrients, concentrations), soil (type, supplier), agar (type, supplier), and cell culture (media, volume, cell number per volume).

  **Growth environment:** This datum includes, but is not limited to, controlled environments such as greenhouse (details on accuracy of control of light, humidity, and temperature), housing conditions (light/dark cycle), and non-controlled environments such as the location of the field trial.

  **Growth time:** This datum refers to the growth time of the sample prior to the treatment.
| Table 11 | Example for the presentation of equipment. | | |
|---|---|---|---|
| example | **Name / manufacturer / model:** *"Inverted confocal microscope, PC and image acquisition software / Zeiss / LSM 780."* | **equipment configuration:** *"Configure a four-channel microscope with appropriate excitation light sources and emission filters: FITC-488 excitation, 490–560-nm emission; ..."* | Protocol available at *Lee et al. (2015)* |

| Table 12 | Reporting consumables. | |
|---|---|---|
| **Ambiguous example** | Filter paper | Protocol available at *Zhang, Nilson & Assmann (2008)* |
| **Descriptive example** | Filter paper (GE, catalog number: 10311611) | Protocol available at *Cao, Zhu & Yan (2014)* |

- **Sample pre-treatment or sample preparation:** This datum refers to collection, transport, storage, preparation (e.g., drying, sieving, grinding, etc.), and preservation of the sample.

*Laboratory equipment.* The laboratory equipment includes apparatus and instruments that are used in diagnostic, surgical, therapeutic, and experimental procedures. In this subsection, all necessary equipment should be listed; manufacturer name or vendor (including the homepage), catalog number (or model), and configuration of the equipment should be part of this data element. See Table 11.

- **Laboratory equipment name:** This datum refers to the name of the equipment as it is given by the manufacturer (e.g., FocalCheck fluorescence microscope test slide).
- **Manufacturer name:** This datum is defined as a person, company, or entity that produces finished goods (e.g., Life Technologies, Zeiss).
- **Laboratory equipment ID (model or catalog number):** This datum refers to an identifier provided by the manufacturer or vendor (e.g., F36909—catalog number for FocalCheck fluorescence microscope test slide from Life Technologies).
- **Equipment configuration:** This datum should explain the configuration of the equipment and the parameters that make it possible to carry out an operation, procedure, or task (e.g., the configuration of an inverted confocal microscope).

*Laboratory consumables or supplies.* The laboratory consumables include, amongst others, disposable pipettes, beakers, funnels, test tubes for accurate and precise measurement, disposable gloves, and face masks for safety in the laboratory. In this subsection, a list with all the consumables necessary to carry out the protocol should be presented with manufacturer name (including the homepage) and catalog number. See Table 12.

- **Laboratory consumable name:** This datum refers to the name of the laboratory consumable as it is given by the manufacturer e.g., Cryogenic Tube, sterile, 1.2 ml.
- **Manufacturer name:** This datum is defined as a person, enterprise, or entity that produces finished goods (e.g., Nalgene, Thermo-scientific, Eppendorf, Falcon)

| Table 13 | Reporting recipes for solutions. | |
|---|---|---|
| Ambiguous example | See in the section recipes, the recipe 1 (PBS) | Protocol available at *Cao, Zhu & Yan (2014)* |
| Descriptive example | Phosphate-buffered saline (PBS) recipe | Protocol available at *Chazotte (2012)* |

- **Laboratory consumable ID (catalog number):** This datum refers to an identifier provided by the manufacturer or vendor; for instance, 5000-0012 (catalog number for Cryogenic Tube, sterile, 1.2 mL from Nalgene).

*Recipe for solutions.* A recipe for solutions is a set of instructions for preparing a particular solution, media, buffer, etc. The recipe for solutions should include the list of all necessary ingredients (chemical compounds, substance, etc.), initial and final concentrations, pH, storage conditions, cautions, and hints. Ready-to-use reagents do not need to be listed in this category; all purchased reagents that require modification (e.g., a dilution or addition of $\beta$-mercaptoethanol) should be listed. See Table 13 for more information.

- **Solution name:** This is the name of the preparation that has at least 2 chemical substances, one of them playing the role of solvent and the other playing the role of solute. If applicable, the name should include the following information: concentration of the solution, final volume and final pH. For instance, Ammonium bicarbonate (NH4HCO3), 50 mM, 10 ml, pH 7.8.
- **Chemical compound name or reagent name:** This is the name of a drug, solvent, chemical, etc.; for instance, agarose, dimethyl sulfoxide (DMSO), phenol, sodium hydroxide. If applicable, a measurable property, e.g., concentration, should be included.
- **Initial concentration of a chemical compound:** This is the first measured concentration of a compound in a substance.
- **Final concentration of chemical compound:** This is the last measured concentration of a compound in a substance.
- **Storage conditions:** This datum includes, among others, shelf life (maximum storage time) and storage temperature for the solutions e.g., "Store the solution at room temperature", "maximum storage time, 6 months". Specify whether or not the solutions must be prepared fresh.
- **Cautions:** Toxic or harmful chemical compounds should be identified by the word 'CAUTION' followed by a brief explanation of the hazard and the precautions that should be taken when handling e.g., "CAUTION: NaOH is a very strong base. Can seriously burn skin and eyes. Wear protective clothing when handling. Make in fume hood".
- **Hints:** The "hints" are commentaries or "tips" that help the researcher to correctly prepare the recipe e.g., "Add NaOH to water to avoid splashing".

*Reagents.* A reagent is a substance used in a chemical reaction to detect, measure, examine, or produce other substances. List all the reagents used when performing the protocol, the

| Table 14 | Reporting reagents. | | |
|---|---|---|
| **Ambiguous example** | Dextran sulfate, Sigma-Aldrich | Protocol available at *Karlgren et al. (2009)* |
| **Descriptive example** | Dextran sulfate sodium salt from *Leuconostoc spp.*, Sigma-Aldrich, D8906-5G | Protocol available at *Javelle, Marco & Timmermans (2011)* |

vendor name (including homepage), and catalog number. Reagents that are purchased ready-to-use should be listed in this section. See Table 14.

- **Reagent name:** This datum refers to the name of the reagent or chemical compound. For instance, "Taq DNA Polymerase from Thermus aquaticus with 10X reaction buffer without MgCl2".
- **Reagent vendor or manufacturer:** This is the person, enterprise, or entity that produces chemical reagents e.g., Sigma-Aldrich.
- **Reagent ID (catalog number):** This is an identifier provided by the manufacturer or vendor. For instance, D4545-250UN (catalog number for Taq DNA Polymerase from Thermus aquaticus with 10X reaction buffer without MgCl2 from Sigma-Aldrich).

*Kits.* A kit is a gear consisting of a set of articles or tools for a specific purpose. List all the kits used when carrying out the protocol, the vendor name (including homepage), and catalog number.

- **Kit name:** This datum refers to the name of the kit as it is given by the manufacturer e.g., Spectrum Plant Total RNA Kit, sufficient for 50 purifications.
- **Kit vendor or manufacturer:** This is the person, enterprise, or entity that produces the kit e.g., Sigma-Aldrich.
- **Kit ID (catalog number):** This is an identifier provided by the manufacturer or vendor e.g., STRN50, catalog number for Spectrum™ Plant Total RNA Kit, sufficient for 50 purifications.

*Software.* Software is composed of a series of instructions that can be interpreted or directly executed by a processing unit. In this subsection, please list software used in the experiment including the version, as well as where to obtain it.

- **Software name:** This datum refers to the name of the software. For instance, "LightCycler 480 Software".
- **Software version:** A software version number is an attribute that represents the version of software e.g., Version 1.5.
- **Software availability:** This datum should indicate where the software can be downloaded from. If possible, license information should also be included; for instance, https://github.com/MRCIE-U/ariesmqtl, GPL3.0.

## Data elements for the procedure

All the analyzed guidelines include recommendations about how to document the instructions; for example, list the steps in numerical order, use active tense, organize
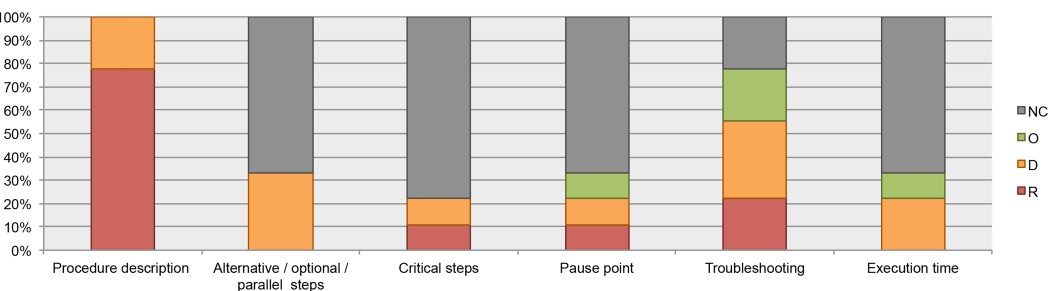

**Figure 6** **Data elements describing the process, as found in the guidelines for authors.** NC, Not Considered in guidelines; O, Optional information; D, Desirable information if this is available; R, Required information.

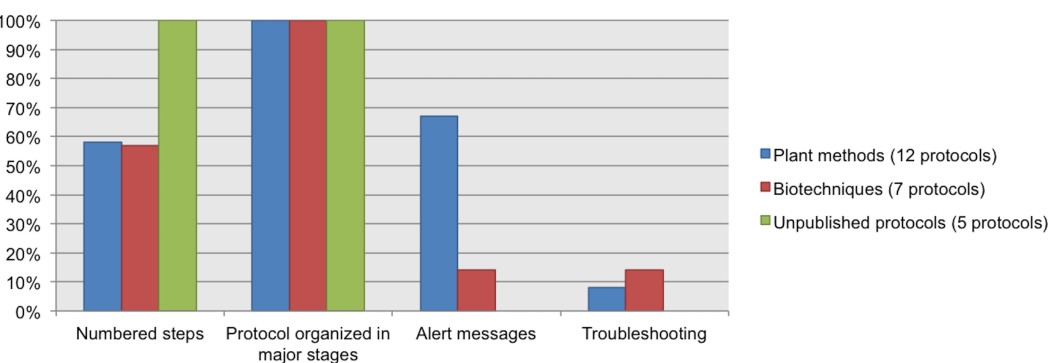

**Figure 7** **Data elements describing the process, as found in analyzed protocols.**

the procedures in major stages, etc. However, information about documentation of alternative, optional, or parallel steps (where applicable) and alert messages such as critical steps, pause point, and execution time was infrequent (available in less than 40% of the guidelines). See Fig. 6.

We chose a subset of protocols (12 from our Plant Methods collection, 7 from our Biotechniques collection, and five unpublished protocols from CIAT) to review which data elements about the procedure were documented. 100% of the protocols have steps organized in major stages. 100% of the unpublished protocols list the steps in numerical order, and nearly 60% of the protocols from Plant Methods and Biotechniques followed this recommendation. Alert messages were included in 67% of the Plant Methods protocols and in 14% of the Biotechniques protocols. Neither of the five unpublished protocols included alert messages. Troubleshooting was reported in just a few protocols; this datum was available in 8% of the Plant Methods protocols and in 14% of the Biotechniques protocols. See Fig. 7.

In this stage, the discussion with domain experts started with the description of steps. In some protocols, the steps are poorly described; for instance, some of them include working temperatures, e.g., cold room, on ice, room temperature; but, *what exactly do they*

*mean?* Steps involving centrifugation, incubation, washing, etc., should specify conditions, e.g., time, temperature, speed (rpm or g), number of washes, etc. For experts, alert messages and troubleshooting (where applicable) complement the description of steps and facilitate a correct execution. This opinion coincides with the results of the survey, where troubleshooting and alert messages such as critical steps, pause points, and timing were considered relevant by 83%–87% of the respondents. The set of data elements related to the procedure is presented below.

- Recommendation 1. Whenever possible, list the steps in numerical order; use active tense. For example: "Pipette 20 ml of buffer A into the flask," as opposed to "20 ml of buffer A are/were pipetted into the flask" (*Nature Protocols, 2012*).
- Recommendation 2. Whenever there are two or more alternatives, these should be numbered as sets of consecutive steps *Wiley's Current Protocols (2012)*. For example: "Choose procedure A (steps 1–10) or procedure B (steps 11–20); then continue with step 21 . . .". Optional steps or steps to be executed in parallel should also be included.
- Recommendation 3. For techniques comprising a number of individual procedures, organize these in the exact order in which they should be executed (*Nature Protocols, 2012*).
- Recommendation 4. Description of steps. Those steps that include working temperatures, e.g., cold room, on ice, room temperature, should be clearly specified. From the European Pharmacopoeia (Pharm.Eur.) (*ECA Foundation, 2017*), World Health Organization resource guidance (WHO guidance) (*WHO, 2003*), and the U.S. Pharmacopeia (USP) (*USP, 2018*), the most common storage conditions were extracted, see below:

  - Frozen/deep-freeze temperature ($-20\,°C$ to $-15\,°C$)
  - Refrigerator, cold room or cold temperature ($2\,°C$ to $8\,°C$)
  - Cool temperature ($8\,°C$ to $15\,°C$)
  - Room/Ambient temperature ($15\,°C$ to $25\,°C$)
  - Warm/Lukewarm temperature ($30\,°C$ to $40\,°C$)

  For centrifugation steps, specify time, temperature, and speed (rpm or g). Always state whether to discard/keep the supernatant/pellet. For incubations, specify time, temperature, and type of incubator. For washes, specify conditions e.g., temperature, washing solution and volume, specific number of washes, etc.

Useful auxiliary information should be included in the form of "alert messages". The goal is to remind or alert the user of a protocol with respect to issues that may arise when executing a step. These messages may cover special tips or hints for performing a step successfully, alternate ways to perform the step, warnings regarding hazardous materials or other safety conditions, time considerations. For instance, pause points, speed at which the step must be performed and storage information (temperature, maximum duration) (*Wiley's Current Protocols, 2012*).

- **Critical steps:** Highlight critical steps in the protocol and give indications that help to carry these out in a precise manner. For instance, time and temperature information

**Table 15  Examples of alert messages.**

| Alert message | Step | Note | Source |
|---|---|---|---|
| Critical step | *"Remove dirt from the surface of the specimen with a tissue. If necessary, moisten the tissue with ..."* | *"Dirt may introduce a variety of inhibitory substances (...); these substances may interfere or even completely block subsequent enzymatic manipulations of the DNA extracts."* | Protocol available at *Rohland & Hofreiter (2007)* |
| Pause point | *"Weigh out no more than 500 mg of sample powder and transfer it to a 15 ml tube."* | *"The sample powder can be stored at room temperature, but should be subjected to the extraction as soon as possible."* | Protocol available at *Rohland & Hofreiter (2007)* |
| Timing | *"Preparation of the bone or tooth sample"* | *"15–30 min per sample"* | Protocol available at *Rohland & Hofreiter (2007)* |
| Hint | *"Add the following components to a nuclease-free microcentrifuge tube:..."* | *"We tested several commercial thermostable DNA polymerases. (...), the most consistent results were obtained using Advantage 2 PCR Polymerase Mix ..."* | Protocol available at *Varkonyi-Gasic et al. (2007)* |

if these are deem crucial. Or, whether the use of RNase free solutions is required. Information should be provided in order to indicate how these steps are critical and how to overcome the issues. ''Critical Steps'' should help the user to maximize the likelihood of success; use the heading CRITICAL STEP followed by a brief explanation. See Table 15.

- **Pause point:** This datum is appropriate after steps in the protocol where the procedure can be stopped. i.e., when the experiment can be stopped and resumed at a later point in time. Any PAUSE POINTS should be indicated with a brief description of the options available. See Table 15.

- **Timing:** This datum is used to include the approximate time of execution of a step or set of steps. Timing could also be indicated at the beginning of the protocol. See Table 15.

- **Hints:** Provide any commentary, note, or hints that will help the researcher to correctly perform the protocol. See Table 15.

- **Troubleshooting:** This datum is used to list common problems, possible causes, and solutions/methods of correction. This can be submitted as a 3-column table or listed in the text. An example is presented in ''Table 1.Troubleshooting table'', available at *Rohland & Hofreiter (2007)*.

## DATA ELEMENTS REPRESENTED IN THE SMART PROTOCOLS ONTOLOGY

The data elements proposed in our guideline are represented in the SMART Protocols Ontology. This ontology was developed to facilitate the semantic representation of experimental protocols. Our ontology reuses the Basic Formal Ontology (BFO) (*IFOMIS, 2018*) and the Relation Ontology (RO) (*Smith et al., 2005*) to characterize concepts. In addition, each term in the SMART Protocols ontology is represented with annotation properties imported from the OBI Minimal metadata. The classes and properties are represented by their respective labels to facilitate the readability; the prefix indicates the

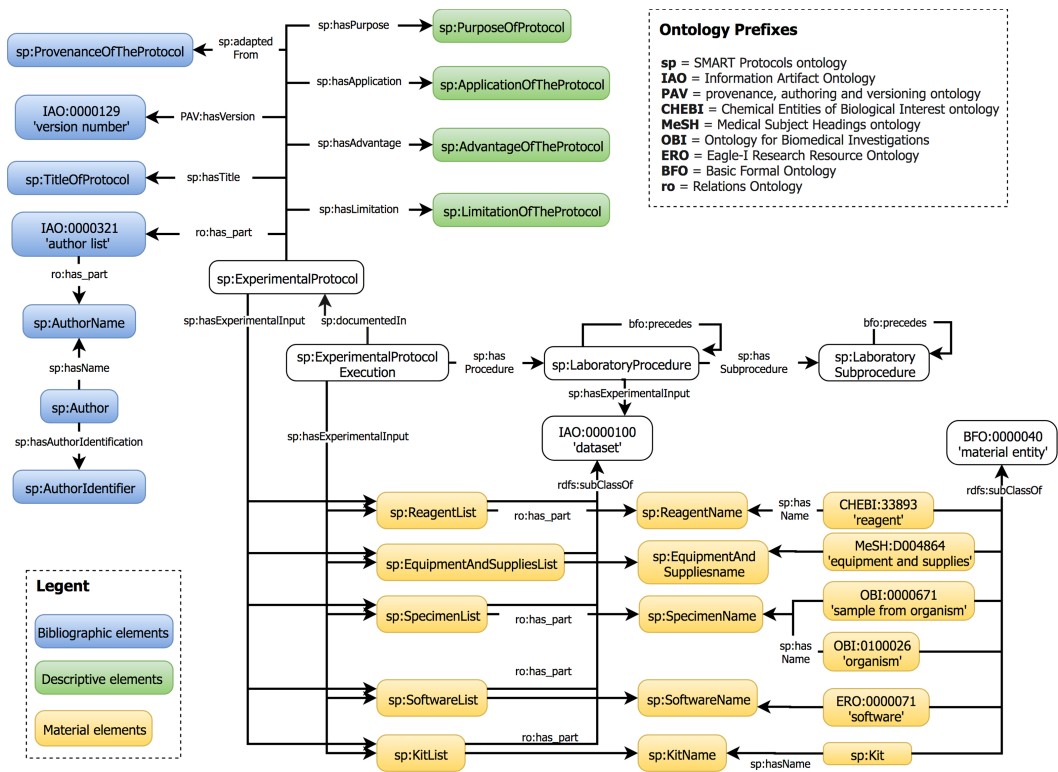

**Figure 8** Hierarchical organization of data elements in the SMART Protocols Ontology.

provenance for each term. Our ontology is organized in two modules. The document module represents the metadata necessary and sufficient for reporting a protocol. The workflow module represents the executable elements of a protocol to be carried out and maintained by humans. Figure 8 presents the hierarchical organization of data elements into the SMART Protocols Ontology.

## DISCUSSION

In this paper, we have described 17 data elements that can be used to improve the reporting structure of protocols. Our work is based on the analysis of 530 published and non-published protocols, guidelines for authors, and suggested reporting structures. We examined guidelines for authors from journals that specialize in publishing experimental protocols, e.g., Bio-protocols, Cold Spring Harbor Protocols, MethodsX, Nature Protocols, and Plant Methods (Methodology). Although JoVE (*JoVE, 2017*) is a video methods journal, its guidelines for authors were also considered. Online repositories were also studied; these resources deliver an innovative approach for the publication of protocols by offering platforms tailored for this kind of document. For instance, protocols.io (*protocols.io, 2018*) structures the protocol by using specific data elements and treats the protocol as a social object, thus facilitating sharing. It also makes it possible to have version control over the document. Protocol Exchange from Nature Protocols is an open repository

where users upload, organize, comment, and share their protocols. Our guideline has also benefited from the input from a group of researchers whose primary interest is having reproducible protocols. By analyzing reporting structures and guidelines for authors, we are contributing to the homogenization of data elements that should be reported as part of experimental protocols. Improving the reporting structure of experimental protocols will add the necessary layer of information that should accompany the data that is currently being deposited into data repositories.

Ours was an iterative development process; drafts were reviewed and analyzed, and then improved versions were produced. This made it easier for us to make effective use of the time that domain experts had available. Working with experimental protocols that were known by our group of domain experts helped us to engage them in the iterations. Also, for the domain experts who worked with us during the workshops, there was a pre-existing interest in standardizing their reporting structures. Reporting guidelines are not an accepted norm in biology (*MIBBI, 2017*); however, experimental protocols are part of the daily activities for most biologists. They are familiar with these documents, the benefits of standardization are easy for them to understand. From our experience at CIAT, once researchers were presented with a standardized format that they could extend and manage with minimal overhead, they adopted it. The early engagement with domain experts in the development process eased the initial adoption; they were familiar with the outcome and aware of the advantages of implementing this practice. However, maintaining the use of the guideline requires more than just availability of the guideline; the long-term use of these instruments requires an institutional policy in data stewardship. Our approach builds upon previous experiences; in our case, the guidelines presented in this paper are a tool that was conceived by researchers as part of their reporting workflow, thus adding a minimal burden on their workload. As domain experts were working with the guideline, they were also gaining familiarity with the Minimum Information for Biological and Biomedical Investigations (MIBBI) (*MIBBI, 2017*) that were applicable to their experiments. This made it possible for us to also discuss the relation between MIBBIs and the content in the experimental protocols.

The quality of the information reported in experimental protocols and methods is a general cause for concern. Poorly described methods generate poorly reproducible research. In a study conducted by *Flórez-Vargas et al. (2014)* in Trypanosoma experiments, they report that none of the investigated articles met all the criteria that should be reported in these kinds of experiments. The study reported by *Kilkenny et al. (2009)* has similar results leading to similar conclusions; key metadata elements are not always reported by researchers. The widespread availability of key metadata elements in ontologies, guidelines, minimal information models, and reporting structures was discussed. These were, from the onset, understood as reusable sources of information. Domain experts understand that they were building on previous experiences; having examples of use is helpful in understanding how to adapt or reuse from existing resources. This helps them to understand the rationale of each data element within the context of their own practice. For us, being able to consult previous experiences was also an advantage. Sharing protocols is a common practice amongst researchers from within the same laboratories or collaborating in the

same experiments or projects. However, there are limitations in sharing protocols, not necessarily related to the lack of reporting standards. They are, for instance, related to patenting and intellectual property issues, as well as to giving away competitive advantages implicit in the method.

During our development process, we considered the SMART Protocols ontology (*Giraldo et al., 2017*); it reuses terminology from OBI, IAO, EXACT, ChEBI, NCBI taxonomy, and other ontologies. Our metadata elements have been mapped to the SMART Protocols ontology; the metadata elements in our guideline could also be mapped to resources on the web such as PubChem (*Kim et al., 2016*) (*Wang et al., 2017*) and the Taxonomy database from UniProt (*UniProt, 2017*). Our implementation of the checklist illustrates how it could be used as an online tool to generate a complement to the metadata that is usually available with published protocols. The content of the protocol does not need to be displayed; key metadata elements are made available together with the standard bibliographic metadata. Laboratories could adapt the online tool to their specific reporting structures. Having a checklist made it easier for the domain experts to validate their protocols. Machine validation is preferable, but such mechanisms require documents to be machine-processable beyond that which our domain experts were able to generate. Domain experts were using the guideline to implement simple Microsoft Word reporting templates. Our checklist does not include aspects inherent to each possible type of experiment such as those available in the MIBBIs; these are based on the minimal common denominator for specific experiments. Both approaches complement each other; where MIBBIs offer specificity, our guideline provides a context that is general enough for facilitating reproducibility and adequate reporting without interfering with records such as those commonly managed by Laboratory Information Management Systems.

In laboratories, experimental protocols are released and periodically undergo revisions until they are released again. These documents follow the publication model put forward by Carole Goble, ''*Don't publish, release*'' with strict versioning, changes, and forks (*Goble, 2017*). Experimental protocols are essentially executable workflows for which identifiers for equipment, reagents, and samples need to be resolved against the Web. The use of unique identifiers can't be underestimated when supporting adequate reporting; identifiers remove ambiguity for key resources and make it possible for software agents to resolve and enrich these entities. The workflows in protocols are mostly followed by humans, but in the future, robots may be executing experiments (*Yachie, Consortium & Natsume, 2017*); it makes sense to investigate other publication paradigms for these documents. The workflow nature of these documents is more suitable for a fully machine-processable or -actionable document. The workflows should be intelligible for humans and processable by machines; thus, facilitating the transition to fully automated laboratory paradigms. Entities and executable elements should be declared and characterized from the onset. The document should be ''born semantic'' and thus inter-operable with the larger web of data. In this way post-publication and linguistic processing activities, such as Named Entity Recognition and annotation, could be more focused.

Currently, when protocols are published, they are treated like any other scientific publication. Little attention is paid to the workflow nature implicit in this kind of

document, or to the chain of provenance indicating where it comes from and how it has changed. The protocol is understood as a text-based narrative instead of a self-descriptive Findable Accessible Interoperable and Reusable (FAIR) (*Wilkinson et al., 2016*) compliant document. There are differences across the examined publications, e.g., JoVE builds the narrative around video, whereas Bio-protocols, MethodsX, Nature Protocols, and Plant Methods primarily rely on a text-based narrative. The protocol is, however, a particular type of publication; it is slightly different from other scientific articles. An experimental protocol is a document that is kept "alive" after it has been published. The protocols are routinely used in laboratory activities, and researchers often improve and adapt them, for instance, by extending the type of samples that can be tested, reducing timing, minimizing the quantity of certain reagents without altering the results, adding new recipes, etc. The issues found in reporting methods probably stem, at least in part, from the current structure of scientific publishing, which is not adequate to effectively communicate complex experimental methods (*Flórez-Vargas et al., 2014*).

## CONCLUSION

Experimental research should be reproducible whenever possible. Having precise descriptions of the protocols is a step in that direction. Our work addresses the problem of adequate reporting for experimental protocols. It builds upon previous work, as well as over an exhaustive analysis of published and unpublished protocols and guidelines for authors. There is value in guidelines because they indicate how to report; having examples of use facilitate how to adapt them. The guideline we present in this paper can be adapted to address the needs of specific communities. Improving reporting structures requires collective efforts from authors, peer reviewers, editors, and funding bodies. There is no "one size that fits all." The improvement will be incremental; as guidelines and minimal information models are presented, they will be evaluated, adapted, and re-deployed.

Authors should be aware of the importance of experimental protocols in the research life-cycle. Experimental protocols ought to be reused and modified, and derivative works are to be expected. This should be considered by authors before publishing their protocols; the terms of use and licenses are the choice of the publisher, but where to publish is the choice of the author. Terms of use and licenses forbidding "reuse", "reproduce", "modify", or "make derivative works based upon" should be avoided. Such restrictions are an impediment to the ability of researchers to use the protocols in their most natural way, which is adapting and reusing them for different purposes –not to mention sharing, which is a common practice among researchers. Protocols represent concrete "know-how" in the biomedical domain. Similarly, publishers should adhere to the principle of encouraging authors to make protocols available, for instance, as preprints or in repositories for protocols or journals. Publishers should enforce the use of repository or journal publishing protocols. Publishers require or encourage data to be available; the same principle should be applied to protocols. Experimental protocols are essential when reproducing or replicating an experiment; data is not contextualized unless the protocols used to derive the data are available.

This work is related to the SMART Protocols project. Ultimately we want: (1) to enable authors to report experimental protocols with necessary and sufficient information that allows others to reproduce an experiment, (2) to ensure that every data item is resolvable against resources in the web of data, and (3) to make the protocols available in RDF, JSON, and HTML as web native objects. We are currently working on a publication platform based on linked data for experimental protocols. Our approach is simple, we consider that protocols should be born semantics and FAIR.

## ACKNOWLEDGEMENTS

Special thanks to the research staff at CIAT; in particular, we want to express our gratitude to those who participated in the workshops, survey and discussions. We also want to thank Melissa Carrion for her useful comments and proof-reading. Finally, we would like to thank the editor and reviewers (Leonid Teytelman, Philippe Rocca-Serra and Tom Gillespie) for their valuable comments and suggestions to improve the manuscript.

### Funding

This work was supported by the EU project Datos4.0 (No. C161046002). Olga Giraldo has been funded by the I+D+i pre doctoral grant from the UPM, and the Predoctoral grant from the I+D+i program from the Universidad Politécnica de Madrid. Alexander Garcia has been funded by the KOPAR project, H2020-MSCA-IF-2014, Grant Agreement No. 655009. The funders had no role in study design, data collection and analysis, decision to publish, or preparation of the manuscript.

### Grant Disclosures

The following grant information was disclosed by the authors:
EU project Datos4.0: C161046002.
UPM.
Universidad Politécnica de Madrid.
KOPAR project: 655009.

### Competing Interests

The authors declare there are no competing interests.

### Author Contributions

- Olga Giraldo conceived and designed the experiments, performed the experiments, analyzed the data, contributed reagents/materials/analysis tools, prepared figures and/or tables, authored or reviewed drafts of the paper, approved the final draft.
- Alexander Garcia contributed reagents/materials/analysis tools, prepared figures and/or tables, authored or reviewed drafts of the paper, approved the final draft, alexander supervised the research and was a constant springboard for discussion and ideas wrt the checklist and methods.
- Oscar Corcho reviewed drafts of the paper, and approved the final draft.

## Data Availability

Federico López Gómez, Alexander Garcia & Olga Giraldo. (2018, March 26). SMARTProtocols/SMARTProtocols.github.io: First release of SMARTProtocols.github.io (Version v1.0.0). Zenodo. http://doi.org/10.5281/zenodo.1207846.

Olga Giraldo. (2018, March 22). oxgiraldo/SMART-Protocols: First release of SMART-Protocols repository (Version v1.0.0). Zenodo. http://doi.org/10.5281/zenodo.1205247.

Olga Giraldo, Alexander Garcia, & Oscar Corcho. (2018). Survey - reporting an experimental protocol [Data set]. Zenodo. http://doi.org/10.5281/zenodo.1204916.

Olga Giraldo, Alexander Garcia, & Oscar Corcho. (2018). Guidelines for reporting experimental protocols [Data set]. Zenodo. http://doi.org/10.5281/zenodo.1204887.

Olga Giraldo, Alexander Garcia, & Oscar Corcho. (2018). Corpus of protocols [Data set]. Zenodo. http://doi.org/10.5281/zenodo.1204838.

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
