# Peer review of "A guideline for reporting experimental protocols in life sciences"

_PeerJ, doi:10.7717/peerj.4795_

## Round 0.1 · original submission · Major Revisions

Decision: Major revisions

The reviews are thorough, constructive and largely in agreement. I have just a few comments to add about specific passages:

1. line 49: "our contribution is extensible and compatible with" other efforts. How would you expect these to be combined in practice?

2. Table 5. How were each of these coded? For instance, does "DOI" indicate "DOIs are the only allowable identifier" or "DOIs plus some other identifiers are allowed" or ... ?

3. Figure 1. Indicate that MI=Minimum Information where it 1st appears in the text

4. line 121: Why are there more MIs reported in Table 3 than in this pgph?

5. Figures 2-8: would prefer to see a y axis label included (even though yes, it can be inferred from the table above)

6. line 176: I'm surprised that 42% is considered high.

7. line 178: "hints" only received 32% support, but was still included. So was there a consistent threshold percentage used to decide what to include, and if so was it decided a priori or not?

8. line 202: How are these strong requirements (e.g. for reporting all author roles) justified?

9. line 219: I'm not sure it is reasonable to require specific versioning conventions here, as opposed to prompting for the version to be tracked in some way

10. line 256: are there any general principle to apply for deciding what about a sample is crucial to report?

In addition, I have made a number of annotations regarding minor issues on the attached PDF.

·

Basic reporting

Please see the "General comments for the author" section.

Experimental design

Please see the "General comments for the author" section.

Validity of the findings

Please see the "General comments for the author" section.

Additional comments

This is a review of “The SMART protocols checklist” manuscript.

The work described here is an extensive undertaking and I would like to genuinely thank the authors for the years of work that they have invested into this. Even though I have spent the last five years learning and thinking about life science protocols, I found the paper to be very valuable and educational. The authors’ approach of studying published protocols, journal guidelines, and expert feedback - while laborious, is precisely what this study needed. The final structural components recommended in the guidelines are all very reasonable and important for reproducibility.

I look forward to the publication of this paper and hope that it can help publishers and editors to improve their reporting guidelines for their journals.

Below is feedback that I hope the authors can use to improve their manuscript.

Kind regards,

Lenny Teytelman
———————————————————————
QUESTIONS AND SUGGESTIONS:

1. What is the intended use of the guidelines?

This paper is a thorough study of what ideally should be in a completely-described protocol. I imagine it will be of interest to the publishers. However, is that the intended audience or is it broader?

Is the hope that authors will follow the proposed guidelines? Is this a first step towards establishing standards and working to get journals to adopt them?

2. Charts in figures

The text in “Discourse data elements” is sufficient to describe the expert responses regarding “objective”, “applications”, “advantages” and “limitations”. Figure 2 carries no extra information and can be removed without loss of clarity. At most, leave the table in Figure 2 (but please change “purpose” to “objective” to be consistent with the text). Same holds for Figures 3,4, 5, 6, 7. Also, please note that in Figure 3, the y-axis instead of starting at zero as in Figure 2 starts at 76% and leads to the erroneous impression that the difference between 15/19 and 16/19 responses is somehow significant.

3. Conclusion

The authors write at the end, “Similarly, publishers should adhere to the principle of encouraging authors to have the protocols available; for instance as preprints or in repositories for protocols or journals. Publishers SHOULD NOT enforce the use of a particular repository or journal.”

I understand the sentiment here and I have a clear conflict of interest in commenting on this as the cofounder and CEO of protocols.io, a repository of research protocols. With that said, I must make the following observations:

• When publishers recommend the deposition of data into repositories, they are often very explicit in their instructions, naming specific repositories that comply with their requirements (e.g. Dryad and figshare).
• If the purpose of the guidelines is to improve manuscripts, compliance by authors is important. Even precise and strict requirements often have very low compliance from the authors (example, data policy from PLOS). Without clear instructions of where and how to deposit protocols, changes to guidelines may become meaningless.
• There are additional considerations when publishers or funders suggest or require the use of specific platforms or resources, even at the cost of liberty to the researchers. For example, specifying GitHub as the platform of choice for code-sharing may be somewhat restrictive. At the same time, if the funder wants to encourage collaboration between the grantees, guiding everyone to the same platform is reasonable.
• Ideally, recommended platforms and repositories should meet a set of requirements (preservation plans, archiving, backups, open APIs, open access, sustainability), but if they do, as I write above, specifying a given resource or resources may be warranted. Imagine if instead of GenBank, the scientists in the Human Genome Project had the freedom to deposit their sequences in any database of their choice. Chromosome I would be in one place, Chromosome 2 in another, and it would be impossible to query all of the genome from a single source.

This manuscript is nicely referenced, but for the argument above, I do not see any prior evidence or analysis that supports the recommendation of the authors. It is critical to balance authors’ freedom with the needs of readers, the barriers to adoption and compliance, and so forth.

———————————————————————
MINOR ISSUES

a) Grammar/spelling:

While the paper is very clear, accessible, and well organized, it does require a careful editing review for grammar. There are many examples of errors such as:

• “Rather that” -> “rather than”
• “the SP checklist aim” -> “the SP checklist aims”
• “shouldn’t researchers be hold” -> “shouldn’t research be held”
• “Methods X” is erroneously “Method X” in references
• “This guideline guideline describes” -> “This guideline describes”
• “the” should be removed in front of %s, as in “the 100% of the domain experts” should just be “100% of the domain experts”.

b) “SMART” looks like an acronym for something but is never defined. Perhaps “structured” or “complete” or “detailed, structured protocols” should be part of the name of the guidelines?

c) NPE is called a journal here; it is not. Protocol Exchange is a repository, similar to protocols.io. They do not perform peer review, but the authors write on page 5, “Analyzing protocols” that the published protocols in their analyses were all peer-reviewed.

d) Instead of “bad” and “good” example on page 10, consider rephrasing as “complete” versus “incomplete” or “detailed” versus ambiguous.

e) On page 10, under Bibliographic data elements, the authors recommend “50 character” or fewer. That seems much too short. Perhaps it’s a mistake and was meant to be “50 words”? The example of a good title in this section actually has 130 characters.

f) When mentioning “protocols.io” on page 11, you may want to cite the following publication: https://doi.org/10.1371/journal.pbio.1002538.

·

Basic reporting

The article address an interest topic in the context of reproducible science.
while the intent is laudable, the overall read could be easier by avoiding repetitions (e.g in the abstract "xperimental protocols are key elements when planning, doing and reporting research. Experimental
protocols are central to research methodologies")

Also transitions between sentences / idea developed are often abrupt or imprecise (e.g. "Experimental protocols are crucial for the execution and reproducibility of any study, the reporting structure remains highly idiosyncratic. However, there is little consistency in the data elements that should be included; the required data elements to be reported within the same discipline vary from publisher to publisher")

The use of 'However' is not suited. So it may be necessary to have the manuscript proof-read by a native speaker to catch these issues.

The manuscript would be greatly improved if the authors could review their draft to really streamline the reading experiment, paying particular attention to the logical transition between sections and thoughts.


The next sentence moves immediately into a specific observation ("Kilkenny et al. (2010a) evaluated 271 journal articles, they found that 4% did not report the number of
animals used anywhere in the methods or the results sections. ") which somehow conflicts with the previous statement in the fact that this reference indicates that 96% of authors did report sample sizes, which seems pretty consistent.

Furthermore, while the authors pick up on how important sample size is, it does not make to final cut of things to report about experimental procedures.
This leads to my next comment :
It would be great to have a delineation of scope for what a protocol is. In clinical context it has a different meaning that in say 'wet lab/molecular biology' context. This has huge implication into what needs to be reported.


Background / state of the art.
The authors may wish to consider efforts such RRID (https://www.force11.org/group/resource-identification-initiative)
More specifically, they could also identify and clearly name major efforts that effectively aim at normalization description of hardware, antibodies and plasmids. The resources are

https://accessgudid.nlm.nih.gov/
www.antibodyregistry.org
www.addgene.org

while RIP (resource information portal) is mentioned, the authors fail to place in context its importantce

Experimental design

Methodology:
The authors should clarify the inclusion criteria for selecting the resource they have analyzed.

In particular, with regard to the semantic artefacts considered, UBERON and NCBI Tax seem quite remote from anything related to describing protocols.
So it seems that section would benefit from distinguishing resources to use as fillers for properties (UBERON,NCBITax,CHEBI) from resources modeling the domain (OBI,BAO,CHMO,EXPO,...)

Then, not much information is available as to what methods were applied to analyse the corpus of data that had been assembled. Was text-mining used? What kind of metrics the authors devised?
For instance, in figure 1 (which is quite nice), a question is 'were protocols described as required by instructions to authors'. Was it done entirely manually ? were there rule based classifiers? was a croud sourcing effort implemented?


But above all, it would be interesting to have the authors discuss the values of 'checklist' in the absence of machine validation?
It would be more important to provide data reporting templates (metadata profiles, configurations)

Validity of the findings

Results:

line194(9), the link 'https://github.com/oxgiraldo/SMART-Protocols/tree/master/SP' returns a 404 but https://github.com/oxgiraldo/SMART-Protocols/ is accessible
and the following seems to work

Why not presenting this table in the main results rather than sending it to the supplementary section?

https://github.com/oxgiraldo/SMART-Protocols/tree/master/SP%20checklist


Looking at the protocol checklist, I am surprised to find a specific item for 'PCR primers'. while those are critical for any amplification based protocol, would it have been to general it to 'nucleic acid', where primers but also oligo-probes, vectors, antisense RNA,....
Why this selection?


While the table is easy to produce and useful, having a machine readable version for instance in the form of JSON schema, with a JSON LD context file which would provide a semantic markup.
It would be a much more modern representation and above all actionable form of the checklist.
Also, it would allow introducing the ontology produced by the authors and which is burried deep into the text.

Additional comments

Discussion:

Putting Biosharing and initiative such as ARRIVE and STAR is incorrect and possibly misleading.
Biosharing is a catalogue of resources about standard terminologies, policies,formats such as ARRIVE or STAR, but does not constitute a recommendation itself. I would encourage the authors to amend to manuscript accordingly (line 408 but also at the begining line 49)

The discussion would benefit from a rewrite as it feels as if many ideas were thrown in but lack clear focus. For instance, @line 432, the authors state "By the same token, our approach complement the ISA tools effort (Sansone et al., 2012)" but it is unclear how? this is where I would find the major weakness in the work presented. it is always oscillating between 'a simple checklist' or a 'structured specification'. This would require addressing.



Conclusion:
Line 456: "Publishers SHOULD NOT enforce the use of a particular repository or journal."
there seems to be a typographic error with 'should not' appearing in bold capital. This makes the whole sentence appear very loud. and as it stands, I would strongly disagree with that statement.

Overall, the message is very unclear as what should be done : recommendation/enforcement. The main reason for heterogeneous reporting is the existence of loopholes, the lack of tools to automatically extract and markup entities and that fact in some cases, (patents) exact protocols and methods can simply not be shared. This is not discussed and may be having a few words would not hurt.



minor:
page 1, line 44: s/founders/funders/
page 16, line 412: s/routinary/routinely

Line 458: Please consider exchanging imprescindible with "essential".

the authors should cite MIBBI as follows:
https://www.ncbi.nlm.nih.gov/pubmed/18688244 or 10.1038/nbt.1411

·

Basic reporting

1) Clear, unambiguous, professional English language used throughout.
The manuscript needs to be reviewed for English. It's not bad, but there are a lot of grammatical errors and non-standard usage.

2) Intro & background to show context:

I don’t think that the introduction sets up the problem appropriately. Both the title ("SMART") and the opening paragraph describes the “data elements” of protocols, implying that the authors are aiming for machine-readable protocols, but that is not what the manuscript is about. Rather, they are laying out essential elements of protocols in the form of a check list. Why is it called “SMART”?

In the discussion, the authors make it clear why protocols are different than a methods section, but that is not made clear in the introduction. Why was a check list approach taken? Why aren’t existing checklists-several of which are mentioned in the introduction-not adequate?

See item 2 in the Experimental design section for suggestions on how to improve the introduction.

3) Literature well referenced & relevant.

Overall yes. Other initiatives and previous approaches to improving protocol reporting like KEfED, and process specification language should be discussed and referenced
.
4) Structure conforms to PeerJ standards, discipline norm, or improved for clarity.

Yes

5) Figures are relevant, high quality, well labelled & described.

Way too many figures, a lot of which are unnecessary. Almost every figure needs a far more complete legend. Figures 2-8 reporting results from a survey of 19 domain experts are interesting but do not further the author's arguments or claims.

The actual Smart protocol elements would be better presented in a single table, rather than 21 separate tables, and should each include a good example and a bad example. The complete table should include things such as the percentage of review protocols that included an 'Application of the protocol' section (for example). More detail can be provided in the text, but the way they are presented now is confusing.

They need to present data about how each part of their checklist is currently represented in the protocols they have reviewed, the author instructions, and the experts.

Table 6: Why is AR relevant? What is the difference between Suggested and Optional? More information needs to be provided.

It looks like some elements, e.g., Description of sample tested are not even considered (NC) in most guidelines. If it isn’t there, how did it end up in this list? As will be detailed in the next section, the methods are under-specified (which is ironic given that this manuscript is about better reporting of methods).

6) Raw data supplied.

The authors provide access to the protocols analyzed via a GitHub repository. Access to data and code is available on github but it needs to be archived via zenodo (or another repository).

The current GitHub repository doesn’t aline exactly with the data presented in Table 2. The Nature Protocols are labeled inconsistently across the two (no mention of Nature in the Git Hub repository). The Molecular Cloning protocols are not referenced in Table 2; the non-polished protocols from CIAT are not in GitHub.

I see a list of instructions to authors, but I don’t see that they have been gathered and made available as a data set. As instructions to authors change, it would be good to create a stable data set from the versions looked at.

A note on references here: shortened URLs are not acceptable for citation as there are no persistence guarantees (eg. line 511, 522). The citation on 522 should be given a zenodo DOI, and cited accordingly.

Experimental design

Within scope:

Yes

2. Research question well defined:
Needs improvement. The research question is clear to a domain expert and is extremely relevant and meaningful but is not sufficiently articulated in the paper. A simple restructuring of the argument of the paper will make it much clearer. Something like the following could work. "Reporting of research protocols is inconsistent. Despite this variability, we wanted to know if there were elements common across protocols. We want to use those elements as the basis for a set of suggestions for reporting protocols. In our analysis we found a common set of elements that we have used to create a practical checklist to help researchers when they report their protocols.”

3. High technical and ethical standard

As the study involved human subjects who were surveyed to produce data, a statement about whether or not this was reviewed by an appropriate IRB board should be included. Although surveys may be exempt, a quick review of on-line materials suggests: “Many survey projects are eligible for exemption.  However, the determination of exempt status (and the type of review that applies) rests with the IRB or with an administration official named by your institution. The determination does not rest with the investigator. “

4. Methods described with sufficient detail.

Needs improvement. The authors describe what was done without providing sufficient detail into how the checklist was actually generated from those steps. The authors repeatedly say that they analysed sources, but it is not at all clear what that entailed. There are a number of black boxes that need to be fleshed out. What were the criteria for including elements on the checklist (where there any)? What were exclusion criteria? Are there examples of elements that were considered but eventually dropped? 18 elements is very large, if you have to reduce the checklist to 10 elements what would they be? The resulting checklist is presented as is, which is fine, but the reasons given for inclusion of certain elements while practical, often do not have any obvious grounding the the various documents that were analyzed. Exact replication is not expected from this kind of work, but the authors need to report more thoroughly the links between their input documents and the elements they propose (see comments on Basic Reporting 4).

Validity of the findings

1. Rationale and benefit to the literature is clearly stated. See item 2 above. Benefit not explicitly stated, but the corpus of documents analyzed and the resulting checklist are major contributions in their own right. But if we have so many of these already and the reporting is still substandard, then why is adding yet another one going to solve the problem?

Validity: It doesn’t look like there was any attempt to balance the types of protocols or the domains covered. The list of topics presented for the protocols does not match the expertise of the domain experts. Also doesn’t look like there was any attempt to validate the results with specific domains that may have not been represented in the expert groups.

2. Statistics:

Needs improvement. No statistics are reported. Given the number of protocols reviewed, it should be possible to give estimates on the portion of the 'population' of protocols that include certain elements, even if only for the domains investigated. The link to the corpus of protocols is not obvious and should be highlighted (it appears as if it were another publication when it is not).

3. Conclusions:.
Needs improvement. The conclusions section is not a conclusion but starts to delve into additional recommendations which either need to be cut, or need to be developed as additional sections. Questions of licensing (as well as patent vs copyright) are critical and if the review of the author instructions shows that no consideration has been made by journals then that is a reportable finding that should be highlighted! While we agree completely that protocols should be required to be reported (they should be in the paper not the supplement! or perhaps published separately and linked), the authors should cite literature related to replication and and reproducibility issues (since direct evidence that poor protocol documentation is the cause of this is still hard to come by) in support of their claims. In addition we would very much contest the idea the the protocol is treated just as any other publication, in the sense that only those protocols that are published are treated as such -- most protocols are not published at all (!) and methods sections do not count. Discussion of any plans to promulgate the checklist for testing whether it improves reproducibility would be much appreciated. Again, if we already have a multiplicity of check lists, guidelines, instructions to authors, ontologies, data models etc., then why is adding yet another one going to help? How will this get adopted?

3. Speculation:

OK

Additional comments

The authors are correct that it is time to treat protocols in a more standard and machine-friendly way and the authors have done a thorough review of the existing practices for reporting protocols in a subset of the life sciences and in that sense is a vital contribution. That said, the paper needs improvement so that it can more clearly communicate how the checklist was developed (specifically a quantification of how the elements included are currently represented across the corpus of protocols), and so that the rationale and motivation are clear to readers who are not also working directly on the problem of protocols. Discussing why this approach is going to help when all others have not is also necessary.


This review was performed by Tom Gillespie, a PhD student working under the supervision of Dr. Maryann Martone. Dr. Martone has read and fully endorses the review.

---

## Round 0.2 · Minor Revisions

I appreciate the effort and care taken in revising the earlier version of this manuscript, but I would ask that you address the remaining comments below. In the response, I would be grateful if you would please quote the passages in the text that address the comments. This will ensure that any substantive idea is captured in the manuscript itself and not just in the response text.

The MAJOR COMMENTS are as follows:

• There is still ambiguity about the meaning of ‘required’ vs. ‘suggested’ and the criteria for deciding that an element belongs in one or the other category, vs not being included at all. Does this choice of words imply something other than the authors intend? Is the distinction between ‘core’ vs. ‘’non-core’ terms, or ones of ‘high importance’ vs. ‘lesser importance’ for reproducibility, or ‘needed to generate valid JSON ‘output’ vs. ‘not needed’, or some combination of these, or something else? If the word ‘required’ is retained, then please make the implied validation or enforcement mechanism more explicit.
• This reflects the larger issue that while the manuscript clearly describes what the inputs (protocols, etc) and outputs (checklists) are, it is still not always clear how you got from one to the other. For example, to say only that the analysis of the corpus was “manual” leaves considerable ambiguity about the steps taken. Given how much the manuscript emphasizes the importance of being comprehensive and accurate in describing a protocol, a reader might well expect this paper to be a model of methodological clarity itself. I realize this comment is not very prescriptive, but I do want you to have the opportunity to review the methods as currently written (along with the detailed comments of Reviewer 4) and revise the text with an eye to greater clarity.
• Please take seriously the comment to use a permanent archive for the supporting data (e.g. a snapshot of the github archive in Zenodo).
• The inclusion of separate Discussion, Perspectives and Conclusion sections is organizationally confusing, since they are all effectively part of the Discussion. I would suggest putting them all under the Discussion heading but, given the length, it would be reasonable to include a few topical subheadings.
• What, if any, licensing conventions were suggested by the guidelines or used in the protocols. Whether or not is addressed, should licensing of the protocol be addressed in the checklist? This was raised in the first round of review and remains to be addressed.
• Table 8 and Figure 10 are lacking legends.
• Table 15 is useful, but it is not clear to me how it will be integrated into the published article, and it would be good to report the classification (i.e. required vs optional, or whatever terms are used there) earlier in the paper. My suggestion (feel free to disagree) would be to include a compact version of this, without the checkmarks, at the beginning of the Results, and to provide the currently formatted version as a supplemental file that can circulate separately from the article.
• Also related to Table 15, I am not familiar with the use of the term ‘marginality’ in this context.

Reviewer 4 has provided much more detailed major comments, but I have tried to extract the essence of what I see as the key issues above, and the authors are not required to respond in more detail to each of Reviewer 4’s comments unless you so choose. In particular, Reviewer 4 raises a number of issues that fall under the umbrella of ‘how does producing this checklist, and in this specific format, relate to the authors’ model of behavioral change among researchers?’. It is my judgement that the authors have addressed this adequately if not fully, and I don’t require them to go further unless they so choose.

In addition, please address the following MINOR COMMENTS:

• line 29: clarify what is meant by 'produced repeatedly' – carried out?
• line 30: 'the descriptions' -> 'their descriptions'
• line 37: 'what' -> 'What'
• multiple cases of 'equipments' -> 'equipment'
• The required elements organism, organism part, and sample/organism identifier in Table 15 are not explicated in the text starting on line 319.
• Modify the sentence starting on line 569 as per Reviewer 4 if that better captures your intended meaning.
• Clarify in the manuscript the restrictions on availability regarding the CIAT protocols.

·

Basic reporting

I am satisfied with the revisions, improvements, and responses in this version.

Experimental design

I am satisfied with the revisions, improvements, and responses in this version.

Validity of the findings

I am satisfied with the revisions, improvements, and responses in this version.

Additional comments

I am satisfied with the revisions, improvements, and responses in this version.

·

Basic reporting

* basic reporting
** Clear, unambiguous, professional English language used throughout.
No, I do not consider the language acceptable. It needs a thorough review.
This has not been addressed.
A subset of things that I spotted.
1. line 17: I do not think 'Life Sciences' is supposed to be capitalized.
2. line 29: 'when produced repeatedly' should read 'when executed repeatedly'? Not sure what the intent was here.
3. line 30: 'the descriptions' -> 'their descriptions'
4. line 37: 'what' -> 'What'
5. line 89: oxford comma 'reviewers, and editors' (there are other cases like this)
6. Spelling error on legend in figure 8.
7. multiple cases of 'equipments' -> 'equipment'
** Intro & background to show context.
1. Despite the authors claim to have rewritten the introduction to discuss the differences between methods sections and protocols I cannot find any evidence of this. lines 570-578 from the discussion should probably be in the introduction
2. There is still no discussion of why the authors have chosen to take a checklist approach. For example, checklists are widely used in safety critical situations such as in airplane cockpits and operation rooms and there are many studies demonstrating their effectiveness. Did this fact or maybe some other fact about checklists have any bearing on the author's choice of format?
3. How is the required subset of the SP Checklist any different from the requirements of minimal information models? If this is an adaptable checklist (line 61) then it should be possible to rank all 51 elements. Furthermore it should be possible to provide advice about the circumstances under which that ranking should change. Regardless of whether the authors call their required subset a minimal information model, it acts as one in the absence of an explanation of how to identify situations where certain elements would not be required.
4. lines 61-65: I agree with the sentiment of these sentences completely, but I think this belongs in the discussion (where it is indeed already discussed). As it stands the authors do not provide citations to support what are very strong claims.
** Literature well referenced & relevant.
1. The authors still do not discuss other approaches to improving the communication of protocols. The authors are describing a tool that can be used to improve protocol reporting. There are other tools that also claim to do this. The point was not that PSL or KeFED in particular 'support' the current work, but rather that the current work needs to be situated in the larger context of other work on protocols. This touches on the fact that the authors have not addressed the question about why they have taken checklist approach from our previous comments.
** Structure conforms to PeerJ standards, discipline norm, or improved for clarity.
Yes.
** Figures are relevant, high quality, well labelled & described.
1. Figure 8 is useful but I think it needs more explanation. I also think that it is probably appropriate to put this in the introduction or as Figure 1 in the results. I can grok it because I maintain an ontology, but I imagine that most readers are going to be hopelessly lost. The crossing black lines are very hard to parse, though I'm not sure anything can be done about it. At the very least a listing of what the curie (qname) prefixes mean is needed. Tangentially related: the ontology seem to be using both an old and new versions of RO and BFO?
2. The simplified figures are much easier to understand. My original comment was a reaction to the information overload evoked by the original figures 2-8.
3. The reduced number of tables is more manageable. I think I may be reacting to the way the current tables are typeset to be wider than the text which breaks up the flow and makes it hard to follow the structure of the sections. Table 15 is great, and I might even put it as Table 1 as context for the rest of the results.
5. Some of the figure captions need more detail. In addition, even if abbreviations are defined once for tables (eg NC, O, R, S) they should probably still be provided again for each figure that uses the abbreviation so that readers do not have to go hunting.
** Raw data supplied (see PeerJ policy).
1. The reasoning about Zenodo is the same reasoning that we used for shortened links. Similarly the author's own rational for including DOIs and not urls from the response to editor comment 2 applies here as well. GitHub is a commercial organization and it does not archive your code, github links are not persistent. GitHub's urls do not have persistence guarantees. Using Zenodo does not duplicate GitHub. It provides a persistent citable link (as a DOI) to a specific version of the repository as it existed at the time of publication so that it is clear to any readers what version of the repo the reviewers saw. Zenodo will also redirect anyone to the live version of the repository and there is no confusion about what the living artifact is (i.e. GitHub). See also: the software citation principles https://doi.org/10.7717/peerj-cs.86. If the authors do not want to use zenodo to provide a persistent citable version of their repository then they should at least link to the commit hash version of the tree so that the provenance can be preserved. For example https://github.com/oxgiraldo/SMART-Protocols/tree/010eba87813652e492461bf8bd14418dc72193b8. Regardless of whether this is in compliance with peerj requirements I as a reviwer cannot claim to have reviewed a GitHub repository because it can change after review. I can claim to have reviewed a GitHub repository at a specific commit.
2. It is fine for the CIAT protocols not to be included, but it needs to be made clear that they are not available as part of the data for this paper since is not clear that that is what 'unpublished' implies in this case. Maybe something to the effect of "75 are unpublished and thus not available in the dataset for this paper" on line 141 would help.

Experimental design

* experimental design
** Original primary research within Scope of the journal.
Yes.
** Research question well defined, relevant & meaningful. It is stated how the research fills an identified knowledge gap.
OK.
** Rigorous investigation performed to a high technical & ethical standard.
OK.
** Methods described with sufficient detail & information to replicate.
1. The methods are clearer but I still think there is a fundamental disconnect between the review of 500+ protocols, the findings as a result that review, and the checklist that the authors propose.

Briefly, all I am asking for is an articulation of the principles that were employed to determine what elements were in the 51 total and what were in the 17 required.

I think that the findings from the review are interesting and important on their own, and I realize that it is out of scope to request the authors to demonstrate that the checklist is effective.
However, the "analysis" mentioned on line 165 is completely opaque.
The methodology for the development of the ontology is well articulated in https://doi.org/10.1186/s13326-017-0160-y. If the current elements for the checklist were derived entirely from the ontology, then simply say so and cite the ontology paper in the methods section. If the elements were all initially derived from the ontology, then it would be nice to know which elements were not considered for inclusion in the checklist. If the answer is that all 16, 17 (the number of R's I count in table 15), 18, or 51, elements were the ones that the experts said should be included that is fine, but I cannot tell if that is the process. How do versions 0.1, 0.2, and 1.0 differ? What elements were added or removed between versions? What was the criteria?

This is partially covered on line 153 "Our checklist includes the data elements that are common across these resources." Are there any elements on the checklist that were not present in all of those resources? If so, what resources were they present in? A version of Table 15 that has one more column which lists the inclusion criteria would be sufficient, even if the criteria was "our experts agreed that this was important to include."

Another concrete example of the problem: why did the number of elements drop from 18 to 16 (17?) between revisions?

Also to this point. I have reviewed a number of protocols myself, though not as many as the authors. I have come to a different set of conclusions about what is required and what is optional. I would very much like to be able to reconcile our choice of elements, but the methodology for inclusion, and the decision to mark as required or suggested is not described.

To give some examples.

For samples and organisms, why is the repository or vendor that the sample procured from not even on the list? Was it considered? This is a serious question, because not only which vendor, but which vendor site animals are procured from is know to have experimentally meaningful effects. (NOTE: organism is missing in the text see Data point 3.) I need to know the starting set of all terms considered for inclusion in order to be able to determine whether the resulting set of elements I am seeing is the result of a decision or because something was simply missed out the outset. This could go in an extended version of table 15 where column one is extended to include all the elements originally considered.

Why is the version of software required if I am using Adobe Photoshop? Does the version have an impact on the outcome of my experiments? Why is version, which in this case has no effect, required when vendor, which is known to have an effect, not even listed for organisms? For Photoshop it does not, but for FreeSurfer it does https://doi.org/10.1371/journal.pone.0038234. "users are discouraged to update to a new major release of either FreeSurfer or operating system without repeating the analysis ... in view of the large and significant cross-version differences ..." Should a biologist cropping an image of a gel have to conform to the needs of the neuroimaging community?

For reagents why are identifiers not required (where they can be obtained)? (Full disclosure: I maintain a tool for crawling RRIDs out of the literature.)

For kits why are the vendor and catalog number not required? They seem like they would be a more reliable way to make sure you have the same "QuickPCR kit."

My point is not that the authors should have done any of the things that I mention above.
My point is that I do not know why they did what they did so I cannot assess why my results are different from theirs.
We are working in different domains, so one major contributing factor might be that the input protocols are somehow different, but I test that unless I know how the authors made their determination.

The inclusion of a particular element is going to depend on (among other things) the criteria that are used to determine whether it is experimentally relevant to the results. The authors seem to prefer human readability over identifiability, but they have not explicitly articulated their criteria, so I do not know for sure.

I am not saying that the authors should promote the use of RRIDs or that they should prefer identifiers over names, but I want to know how they made the decision, otherwise I cannot reconcile my results with theirs.

2. With regard to the number of elements. The point was not that 10 is better than any other number or that the authors should be writing a certain kind of paper. The point is that the authors should be able to provide a ranked list of their elements from most important to least important. In context, the question was posed as a journal who was interested in adopting this checklist but only has room for 10 elements (or 5, or 6, or 12 elements, etc.). If the authors can only tell the journal "well you have to use all 18" then that is a lost opportunity.

Validity of the findings

* validity of findings
1. A note on our intention with asking for an explanation of why this checklist is useful. First, that question came from Maryann, so I can only speak as a proxy here. That said, it is not that we have anything against checklists, or any other tool in particular. It is more that there are so many tools out there already and we still haven't solved the problem. Is the issue that we don't have the right tool? Or is the issue that the tools we have, have not been supported/promulgated after release/publication? This is also the reason for the later question about promulgation.

I cannot speak to the question about mismatch between experts and protocols since that one came from Maryann. In the event that readers might raise the same question, maybe a list of the domains in which the experts were expert like the one provided in the rebuttal would help?

I can however, state that the lack of cross validation in other domains is still a concern. My question about why vendor is not listed as a required element for organisms is an example of this. In some domains the protocol will need to be that specific, in others it will not.

We also very much agree that more work and effort on a variety of solutions for protocols is needed. The reason we brought up those questions is because we personally have received them from experimentalists. We were suggesting that it is important to make the value of the checklists explicit in the paper as the authors have done in their response to our comments.
** Data is robust, statistically sound, & controlled.
1. Stats OK.
2. The suggestions from table 15 for organism/sample do not match what is in the text. All 3 required elements organism, organism part, and Sample/organism identifier, are not explicated in the text starting on line 319.
3. I don't know whether this is an issue for the authors, but table 14 should come before the bullets starting on line 480 so as not to break up the list.

** Conclusions are well stated, linked to original research question & limited to supporting results.
1. With regard to licensing. If you have reviewed the author instructions for numerous journals and they do not say anything about how the protocol will be licensed then that is an important finding.
2. We concur with the authors in that we have been unable to find any work directly addressing the relationship between protocol quality and reproducibility beyond the ones that they cite.
3. I would still like the sentence starting on line 569 to be explicitly qualified because I think that it can be misread to imply that protocols are routinely published. This is clearly not what the authors intend. Something to the effect of "Currently, when protocols are published, they are treated like any other scientific publication, as if they are just text." The additional implication being that methods sections certainly are not FAIR and that even when published independently protocols are still not FAIR, as the authors discuss elsewhere.
4. We are not asking for a continuation of the study, we are asking for a sentence or two describing plans for continuation if there are any.
5. We do not consider it useless to publish checklists. We are asking the authors to consider that there are many people that have reached 'new tool fatigue' and who are likely to dismiss efforts like this out of hand, especially if the authors do not articulate how their approach is any different. How does this make the lives of experimenters easier? I will echo the question about wanting evidence for the sentence on line 583-584, and having that connected to how this approach is different. We are not discouraging the work at all, we are asking the authors to articulate why they themselves are not discouraged given the repeated failures of efforts that on the surface appear to be similar to theirs. For example, checklists could be easier to adopt than ontologies or minimal information models. There is no expectation that the authors solve the promulgation problem, which is one of the most fundamental problems period. There is however, a desire to know whether the authors have considered anything beyond the publication of this paper.
** Speculation is welcome, but should be identified as such.
1. line 516: 'can be used' -> 'could be used'. The authors provide no evidence about the efficacy of their checklist.

Additional comments

I think that this paper is really two papers. This first is about the findings you have made about the current practices for reporting protocols in the literature and among domain experts. You have lots of data here that is sufficient for a single publication. The second is about the checklist. Currently you have no data about the checklist or whether or not it is effective. That is fine if you articulate the criteria used to select the elements of the checklist, but it is still hard to defend. Having more evidence about the efficacy of the checklist will make it much easier to publish the checklist itself.

---

## Round 0.3 · accepted · Accept

Thank you for responding to multiple round of detailed reviews, and I hope you feel, like me, that the manuscript is the better for it.

#